# DIFFERENTIALLY PRIVATE GEODESIC REGRESSION

## ABSTRACT

In statistical applications it has become increasingly common to encounter data structures that live on non-linear spaces such as manifolds. Classical linear regression, one of the most fundamental methodologies of statistical learning, captures the relationship between an independent variable and a response variable which both are assumed to live in Euclidean space. Thus, geodesic regression emerged as an extension where the response variable lives on a Riemannian manifold. The parameters of geodesic regression, as with linear regression, capture the relationship of sensitive data and hence one should consider the privacy protection practices of said parameters. We consider releasing Differentially Private (DP) parameters of geodesic regression via the K-Norm Gradient (KNG) mechanism for Riemannian manifolds. We derive theoretical bounds for the sensitivity of the parameters showing they are tied to their respective Jacobi fields and hence the curvature of the space. This corroborates, and extends, recent findings of differential privacy for the Fréchet mean. We demonstrate the efficacy of our methodology on the sphere, $S_2 \subset \mathbb{R}^3$, the space of symmetric positive definite matrices, and Kendall's planar shape space. Our methodology is general to any Riemannian manifold and thus it is suitable for data in domains such as medical imaging and computer vision.

## 1 INTRODUCTION AND MOTIVATION

One of the most foundational tools in statistical analyses is linear regression. In its simplest form, linear regression learns a linear relationship between an independent variable, the predictor, and a dependent variable, the response. Typically, these variables are both assumed to lie in a flat, Euclidean space. However, in modern statistical practices, it is common to encounter data that inherently live in curved, non-Euclidean spaces such as spherical data (e.g., directional wind data, spatial data, square-root discrete distributions) (Fisher et al., 1993; Jeong et al., 2017), symmetric positive definite matrices (e.g., covariance matrices, brain tensors) (Fletcher and Joshi, 2004). For this reason, there have been many different extensions of regression to non-linear spaces; Faraway (2014) considered regression on metric spaces only requiring pairwise distances between the observations, Wasserstein Regression (Chen et al., 2023) captures the relationship between univariate distributions as predictors and either distributions or scalars as the response, Fréchet Regression (Petersen and Müller, 2019) captures the relationship between Euclidean predictors and responses that lie in a metric space, and Geodesic Regression (Fletcher, 2011) captures the relationship between Euclidean predictors and response variables that lie on a Riemannian manifold, for instance.

Whatever space the data may live, the learned relationship between variables relies on data captured from individuals; these individuals may be concerned with safeguarding their sensitive data. To protect one's data, differential privacy (DP, Dwork et al., 2006) has emerged as a leading standard for data sanitisation. Perhaps surprisingly, differential privacy for a methodology as fundamental as linear regression is not so straightforward. A common approach for private linear regression involves sanitizing the matrix of sufficient statistics, $A^T A$, where $A$ is the matrix of data with rows being observations and columns being features. This is explored by Sheffet (2019) and Dwork et al. (2014), although the latter considered this in the case of private PCA. Further, Wang (2018), Sheffet (2017), and Alabi et al. (2020) survey the landscape of techniques for private linear regression as well as propose their own algorithms, so all this is to say that the fundamental and seemingly simple task is not trivial.

Recently, there has been an emergence of extensions of differential privacy techniques into Riemannian manifolds; Reimherr et al. (2021) considered privately estimating the Fréchet mean on

Riemannian manifolds and, in the process thereof, extended the Laplace mechanism. They note that the sensitivity of the Fréchet mean is tied to the curvature of space and inflated for positively curved manifolds. They further note that sanitising on the manifold incurs less noise than sanitising in an ambient space under the same privacy budget. Following this work, mechanism design for manifolds has been developed such as the K-Norm Gradient (KNG) mechanism for manifolds (Soto et al., 2022), DP Riemannian optimisation (Han et al., 2024a;b), and extensions of DP definitions such as Gaussian DP for manifolds (Jiang et al., 2023).

In this paper, we consider the problem of privately estimating the parameters of geodesic regression (Fletcher, 2011), regression with a Euclidean predictor and a response variable on a Riemannian manifold. Geodesic regression is parametrised by a footpoint, an initial value on the manifold, and a shooting vector, the average direction of data; our method sanitises the parameters sequentially with the Riemannian manifold extension of the KNG mechanism (Reimherr and Awan, 2019; Soto et al., 2022). We show that the sensitivity of each parameter is tied to their respective Jacobi field equations and hence the curvature of the manifold. We demonstrate our methodology on the sphere, the space of symmetric positive definite matrices, and Kendall's planar shape space (Kendall, 1984). Our methodology, however, is general to any Riemannian manifold and simply requires an understanding of the Jacobi fields which are specific to the manifold of interest.

## 2 BACKGROUND AND NOTATION

### 2.1 RIEMANNIAN MANIFOLDS

In the following, we summarise the necessary tools of differential geometry for handling Riemannian manifolds. Some additional details can be found in B and we refer to classical texts such as Do Carmo (1992) and Nakahara (2018) for a more thorough introduction.

A manifold $\mathcal{M}$ is a topological space that is locally equivalent to Euclidean space $\mathbb{R}^n$. A manifold can be further endowed with a metric giving it additional structure such as the ability to measure angles and lengths. This is a natural generalization of the inner product between vectors in $\mathbb{R}^n$ to an arbitrary manifold. A Riemmanian metric $g : T_p\mathcal{M} \mapsto \mathbb{R}$ is a $(0, 2)$ tensor field on $\mathcal{M}$ that is symmetric $g_p(U, V) = g_p(V, U)$ and positive definite $g_p(U, U) \geq 0$ where $U, V$ lie on the tangent space of $\mathcal{M}$ at $p$ denoted by $T_p\mathcal{M}$. A smooth manifold that admits a Riemannian metric is called a Riemannian manifold.

We can define an exponential map at point $p \in \mathcal{M}$ as $\text{Exp}(p, v) := \gamma_v(1)$ where $\gamma : [0, 1] \to \mathcal{M}$ is a geodesic. The exponential map, maps a vector $v \in T_p\mathcal{M}$ to $\gamma(1) \in \mathcal{M}$. Similarly, we can define the inverse of the exponential map, the log map, $\text{Log}(p, q) : q \in P \to T_p\mathcal{M}$ where $P \subset \mathcal{M}$ is the largest normal neighborhood of $p$. We use parallel transport to compare and combine vectors on different tangent spaces of a manifold. Parallel transport provides a way to move tangent vectors along a curve while preserving their length from one tangent space to another. Given a smooth curve $\gamma(t)$ starting at point $p$, parallel transport ensures that the vector $v(t)$, $v \in T_p\mathcal{M}$, remains 'straight' relative to the manifold's curvature as it moves along the curve $\gamma(t)$ by satisfying $\nabla_{\dot{\gamma}(t)} v(t) = 0$. We denote $\Gamma_p^q v$ as the parallel transport of vector $v \in T_p\mathcal{M}$ to the tangent space $T_q\mathcal{M}$. Since parallel transport moves tangent vectors along smooth paths in a parallel sense, it preserves the Riemannian metric and hence angles between vectors. That is, for a smooth curve $\gamma : [0, 1] \to \mathcal{M}$ and tangent vectors $v_1, v_2 \in T_{\gamma(0)}\mathcal{M}$ we have that $\langle v_1, v_2 \rangle_{\gamma(0)} = \left\langle \Gamma_{\gamma(0)}^{\gamma(1)} v_1, \Gamma_{\gamma(0)}^{\gamma(1)} v_2 \right\rangle_{\gamma(1)}$.

Geodesics on a manifold are affected by the curvature of the manifold. Roughly, positive curvature causes geodesics to converge and negative curvature causes geodesics to diverge. Jacobi equations are a way of quantifying this dependence of curvature on the geodesic. A vector field satisfying the Jacobi equation is called a Jacobi field. For a geodesic $\gamma$ and a vector field $J$ along $\gamma$, the Jacobi equation is defined as,

$$\frac{D^2}{dt^2} J(t) + R\left(J(t), \frac{d}{dt}\gamma\right)\frac{d}{dt}\gamma = 0, \tag{1}$$

where $R$ is the Riemannian curvature tensor, see B.2. One should note that the Riemannian curvature is the more general form of sectional curvature which is used later. An important result of Riemannian geometry that will be useful later is the Rauch theorem, which states:

**Theorem 2.1** (Rauch Comparison Theorem). *For two Riemannian manifolds* $\mathcal{M}, \tilde{\mathcal{M}}$ *with curvatures* $K(\gamma), \tilde{K}(\tilde{\gamma})$ *and geodesics* $\gamma : [0, \beta] \rightarrow \mathcal{M}$, $\tilde{\gamma} : [0, \beta] \rightarrow \tilde{\mathcal{M}}$ *let* $J, \tilde{J}$ *be the Jacobi fields along* $\gamma, \tilde{\gamma}$, *respectively. If* $K(\gamma) \leq \tilde{K}(\tilde{\gamma})$ *then,*

$$\|\tilde{J}\| \leq \|J\|. \tag{2}$$

Intuitively, this states that for large curvature geodesics tend to converge, while for small (or negative) curvature geodesics tend to spread. As curvature increases, lengths shorten.

## 2.2 GEODESIC REGRESSION

Geodesic regression provides a framework for modeling the relationship between a real-valued independent variable and a manifold-valued dependent variable, leveraging the intrinsic geometry of Riemannian manifolds. Unlike traditional linear regression, geodesic regression generalizes the concept to non-linear spaces, representing relationships as geodesic curves on the manifold. Fletcher (2011) formulated the least-squares estimation of geodesic regression by minimizing the sum of squared geodesic distances between data points and the estimated prediction geodesic. Consider a dataset $D = \{(x_i, y_i)\}$ where $(x_i, y_i) \in \mathbb{R} \times \mathcal{M}$, for $i = 1, ... n$. Here $x_i$ lie on the real line and can be scaled to be between $[0, 1]$ and $y_i$ lie on a Riemannian manifold $\mathcal{M}$. To estimate the regression parameters, $(p, v) \in T\mathcal{M}$ we need to minimise the least squared energy given by:

$$E(p, v) = \frac{1}{2\,n} \sum_{i=1}^{n} d(\mathrm{Exp}(p, x_i\, v), y_i)^2, \tag{3}$$

$$(\hat{p}, \hat{v}) = \mathrm{argmin}_{(p, v)}\ E(p, v). \tag{4}$$

We denote the parameters that minimise the least square energy 3 as $\hat{p}, \hat{v}$. Here, $d(\cdot, \cdot)$ is the geodesic distance on the manifold, so the energy is the sum of square distances from the predicted value, $\mathrm{Exp}(p, x_i v)$, and the observed value, $y_i$. This framework enables parametrisation through an initial point, or footpoint, and velocity, or shooting vector. This parametrisation is analogous to an intercept and slope in linear regression. The gradient of the energy with respect to the footpoint ($p$) and the shooting vector ($v$) can be found using properties of the Riemannian gradient (refer to B.3 for further details):

$$\nabla_p E = -\frac{1}{n} \sum_{i=1}^{n} d_p\mathrm{Exp}(p, x_i\, v)^{\dagger}\, \varepsilon_i, \tag{5}$$

$$\nabla_v E = -\frac{1}{n} \sum_{i=1}^{n} x_i\, d_v\mathrm{Exp}(p, x_i\, v)^{\dagger}\, \varepsilon_i, \tag{6}$$

respectively, where the errors are defined as $\varepsilon_i = \mathrm{Log}(\mathrm{Exp}(p, x_i\, v), y_i)$ and $\dagger$ is the adjoint operator defined as $\langle Au, w \rangle_q = \langle u, A^{\dagger}w \rangle$ for $A : T_p\mathcal{M} \rightarrow T_q\mathcal{M}, u \in T_p\mathcal{M}, w \in T_q\mathcal{M}$. Further, for existence and uniqueness of the optimal solutions, only geodesics "close" to the data are considered, where a geodesic $\gamma$ is defined as $\tau-$close if $d(\gamma(x_i), y_i) \leq \tau$ for all $i$, $\tau > 0$, and $\gamma$ does not pass through the cut locus of the manifold. One should note that before performing the regression task we scale our covariates ($x_i$) using an affine transformation such that $x_i \in [0, 1]$ with some $x_k = 0$ and another $x_{k'} = 1$.

As opposed to linear regression, where the errors are scalar values, here the errors are vectors in the tangent spaces of the predicted values, $T_{\mathrm{Exp}(p, x_i v)}\mathcal{M}$. The derivative of the exponential map with respect to the footpoint $p$ can calculated by varying $p$ along the geodesic $\eta(s) = \mathrm{Exp}(p, s\, u_1)$, where $u_1 \in T_p\mathcal{M}$. This will result in the variation of the geodesic given by $c_1(s, t) = \mathrm{Exp}(\mathrm{Exp}(p, s\, u_1), t\, v(s))$. Similarly, the derivative of the exponential map with respect to the shooting vector $v$ is found by a varying $v$ resulting in the variation of geodesic $c_2(s, t) = \mathrm{Exp}(p, s\, u_2 + t\, v)$, where $u_2 \in T_p\mathcal{M}$. This gives the derivatives of the exponential map as:

$$d_p \mathrm{Exp}(p, v)\, u_1 = \partial_s c_1(0, 1) = J(1), \qquad J(0) = u_1,\ J'(0) = 0 \tag{7}$$

$$d_v \mathrm{Exp}(p, v)\, u_2 = \partial_s c_2(0, 1) = J(1), \qquad J(0) = 0,\ J'(0) = u_2 \tag{8}$$

where $J_i(t)$ are Jacobi fields along the geodesic $\gamma(t) = \mathrm{Exp}(p, t\, v)$.

## 2.3 DIFFERENTIAL PRIVACY

In the modern era of data collection, the protection of privacy of one's data has become increasingly prevalent. There is no one consensus on what one means by *privacy*. However, *differential privacy* (Dwork et al., 2006) has recently emerged as a defacto definition. Since its initial definition there have been many extensions and alternate definitions such as concentrated DP (Dwork and Rothblum, 2016), zero-concentrated DP (zCDP) (Bun and Steinke, 2016), Rényi differential privacy (Mironov, 2017), and Gaussian differential privacy ($\mu-$GDP) (Dong et al., 2022), for instance. Even though there are different forms of differential privacy, all definitions rely on the idea of *adjacent* datasets.

Let $D = \{(x_1, y_1), \ldots, (x_n, y_n)\} \subset (\mathbb{R} \times \mathcal{M})$ denote a dataset of size $n$. An adjacent dataset $D'$ differs from $D$ in exactly one record, or observation, which we can take, without loss of generality, to be the last i.e., $D' = \{(x_1, y_1), \ldots, (x_n', y_n')\}$. Adjacent datasets are denoted as $D \sim D'$.

**Definition 2.2.** A randomised mechanism $f(z; D)$ is said to satisfy *pure differential privacy* if

$$P(f(z; D) \in A) \leq \exp(\epsilon) P(f(z; D') \in A),$$

for given privacy budget $\epsilon > 0$, all $D \sim D'$, and $A$ is any measurable set in $\mathcal{M}$.

Roughly speaking, for small $\epsilon$, a mechanism that satisfies pure differential privacy is approximately equally likely to observe a realization from the random mechanism over all adjacent datasets. This definition of privacy is attractive as it ensures noise calibration relative to an individual's effect on the mechanism relative to the entire dataset. Differential privacy is well defined over Riemannian manifolds via the Riemannian measure (Reimherr et al., 2021) as it is well defined over measurable spaces (Wasserman and Zhou, 2010). An attractive property of this definition is the composition of mechanisms. Given two mechanisms $f_1, f_2$ with privacy budgets $\epsilon_p, \epsilon_v$, respectively, the total privacy budget is $\epsilon_p + \epsilon_v$. Note these $\epsilon$ are privacy budgets and not the errors $\varepsilon$ from 2.2.

We wish to release private versions of the parameters $(p, v)$, the footpoint, and the shooting vector of geodesic regression, respectively. Generally speaking, these parameters are not available in closed form but rather are optimisers of an energy function. As such, it is useful to consider the exponential mechanism (McSherry and Talwar, 2007). The exponential mechanism is a randomized mechanism that releases values nearly optimising an energy function. Generally, it takes the form

$$f(z; D) \propto \exp\{-\sigma^{-1} E(z; D)\},$$

where $\sigma$ is a spread parameter which determines the noise scale, $E(z; D)$ is an energy function to be minimised, and $z$ is the random variable. A particular instantiation of the exponential mechanism is the K-Norm Gradient mechanism (KNG, Reimherr and Awan, 2019). KNG has been shown to have statistical utility gains in Euclidean spaces as compared to the exponential mechanism. This mechanism was extended to Riemannian manifolds (Soto et al., 2022) and shown to have similar utility gains as in the Euclidean case.

In general, the KNG mechanism on manifolds takes the form

$$f(z; D) \propto \exp\{-\sigma^{-1} \|\nabla E(z; D)\|_z\}. \tag{9}$$

KNG satisfies pure differential privacy when the noise scale is given by $\sigma = \Delta/\epsilon$ where $\Delta = \sup_{D \sim D'} \|\nabla E(z; D) - \nabla E(z; D')\|_z$ is the global sensitivity and $\epsilon$ is the privacy budget or $\sigma = 2\Delta/\epsilon$ if the normalizing constant is dependent on the footpoint of the mechanism (Soto et al., 2022). We further note that the sensitivity needs to be determined for ones choice of energy function.

## 3 SENSITIVITY ANALYSIS

As discussed earlier we use the KNG mechanism to sanitise the footpoint and the shooting vector. For KNG to satisfy pure DP the sensitivity $\Delta = \sup_{D \sim D'} \|\nabla E(z; D) - \nabla E(z; D')\|_z$ must be bounded. Since we estimate two parameters, each parameter has its own sensitivity bound. We let $\Delta_p, \Delta_v$ be the sensitivity of the footpoint and shooting vector, respectively. In this section we consider the sensitivity of the footpoint $\Delta_p$ and in D we consider the sensitivity of the shooting vector.

We wish to release a private version of $\hat{p}$ which is an optimiser of the energy function $E(p, v; D)$ as in equation 3. Since $E(p, v; D)$ is a function of both $p$ and $v$ we must, in a sense, consider

them separately and sequentially as the gradient for KNG can be taken with respect to either parameter. One should note that our sequential treatment of sensitivities does not rely on any global product decomposition of the tangent bundle $T\mathcal{M}$. Recall that $T\mathcal{M}$ is a fiber bundle with projection $\pi : T\mathcal{M} \to \mathcal{M}$, whose fiber over $p \in \mathcal{M}$ is the tangent space $T_p\mathcal{M}$. In our procedure, fixing $p$ selects a base point of $\mathcal{M}$, which determines the corresponding fiber $T_p\mathcal{M}$; the vector $v$ is then sampled from this fiber. Thus, the process of first sampling $p$ and subsequently $v$ makes use only of the canonical fiber-bundle structure of $T\mathcal{M}$ and does not assume a global product structure $T\mathcal{M} \cong \mathcal{M} \times \mathbb{R}^n$.

For existence, uniqueness, and a finite sensitivity, we introduce the following assumptions.

**Assumption 3.1.** *There exist constants $\kappa_l, \kappa_h \in \mathbb{R}$ such that the sectional curvature at every point of the manifold, with respect to every $2$-plane in the tangent space, lies in the interval $(\kappa_l, \kappa_h)$.*

**Assumption 3.2.** *For dataset $D \in \mathcal{D}$, the data is bounded as $D \subseteq B_r(m_0)$ where $r \leq \frac{\pi}{8\sqrt{\kappa_h}}$ for Riemannian manifolds with positive curvature and $r < \tau_m$ for Riemannian manifolds with negative curvature. Further the least-squares geodesic is $\tau-$close to the data. For $\tau > 0$, $\sup_D d(y_i, Exp(\hat{p}, x_i\hat{v})) \leq \tau, \forall i$.*

**Theorem 3.3.** *Let Assumptions 3.1 and 3.2 hold. Let $D, D'$ be adjacent datasets, for a fixed shooting vector $v$,*

$$\Delta_p \leq \begin{cases} \dfrac{2\tau}{n} & , \kappa_l \geq 0, \\ \dfrac{2\tau}{n}\cosh(2\sqrt{-\kappa_l}(\tau_m + \tau)) & , \kappa_l < 0. \end{cases}$$

*Proof.* We will start with fixing shooting vector, $v$ and focus on the footpoint, $p$. The global sensitivity is

$$\Delta_p = \sup_{D \sim D'} \|\nabla_p E(p; D) - \nabla_p E(p; D')\|. \tag{10}$$

Using equation 5 the gradient of the energy is given by, $\nabla_p E(p; D) = -\frac{1}{n}\sum_{i=1}^{n} d_p\text{Exp}(p, x_i v)^\dagger \vec{\varepsilon_i}$ where $\vec{\varepsilon_i} = \text{Log}(\text{Exp}(p, x_i v), y_i)$ is the error vector and $\dagger$ denotes the adjoint operator. The norm in equation 10 is thus the difference of two sums that differ in only one term due to the adjacent datasets $D \sim D'$. All terms, in each gradient term, thus, cancel except the last with covariates $x_n, x'_n$ and respective errors $\varepsilon_n, \varepsilon'_n$ from $D$ and $D'$. We have,

$$\Delta_p = \frac{1}{n}\|d_p\text{Exp}(p, x_n v)^\dagger \vec{\varepsilon_n} - d_p\text{Exp}(p, x'_n v)^\dagger \vec{\varepsilon_n}'\|, \tag{11}$$

$$\leq \frac{1}{n}\left(\|d_p\text{Exp}(p, x_n v)^\dagger \vec{\varepsilon_n}\| + \|d_p\text{Exp}(p, x'_n v)^\dagger \vec{\varepsilon_n}'\|\right), \tag{12}$$

$$\leq \frac{1}{n}\left(\|d_p\text{Exp}(p, x_n v)\|_{op}\|\vec{\varepsilon_n}\| + \|d_p\text{Exp}(p, x'_n v)\|_{op}\|\vec{\varepsilon_n}'\|\right), \tag{13}$$

$$\leq \frac{\tau}{n}\left(\sup_{\|\vec{u}\|=1}\|J_{\vec{u}}^{(x_n)}(1)\| + \sup_{\|\vec{u}\|=1}\|J_{\vec{u}}^{(x'_n)}(1)\|\right). \tag{14}$$

Here the second step applies the triangle inequality, the third step uses the characterization $\|A\vec{\varepsilon_n}\| \leq \|A\|_{op}\|\vec{\varepsilon_n}\|$ for linear operators, and the operator norm is preserved under the adjoint, i.e. $\|A^\dagger\| = \|A\|$. In the final step, using the definition of the operator norm and the fact that $d_p\text{Exp}(p, x_n v)[\vec{u}] = J_{\vec{u}}^{(x_n)}(1)$ for the Jacobi field along $\gamma(t) = \text{Exp}(p, t\,x_n v)$ with initial conditions $J_{\vec{u}}^{(x_n)}(0) = \vec{u}$ and $J_{\vec{u}}'^{(x_n)}(0) = 0$, we obtain $\|d_p\text{Exp}(p, x_n v)\|_{op} = \sup_{\|\vec{u}\|=1}\|J_{\vec{u}}^{(x_n)}(1)\|$. According to Assumption 3.2, the errors $(\varepsilon_n, \varepsilon'_n)$ are bounded in norm by $\tau$. We therefore set $\|\varepsilon_n\| = \|\varepsilon'_n\| = \tau$ in the last equation to obtain the tightest bound.

Next, let's consider only the first part of equation 14. Using the Rauch comparison theorem by taking the model manifold $\tilde{\mathcal{M}}$ with constant sectional curvature $\kappa = \kappa_l$ we get $\sup_{\|\vec{u}\|=1}\|J_{\vec{u}}^{(x_n)}(1)\| \leq \sup_{\|\vec{u}\|=1}\|\tilde{J}_{\vec{u}}^{(x_n)}(1)\|$ where $\tilde{J}$ is the Jacobi field on our model manifold $\tilde{\mathcal{M}}$.

We can next decompose $\vec{u}$ into perpendicular and parallel components to the geodesic. For constant curvature manifolds, $\|\tilde{J}_{||}(1)\| = \|\vec{u}_{||}\|$ and $\tilde{J}_\perp$ is dependent on the curvature, $\|\tilde{J}_\perp(1)\| =$

$\|C_{\kappa_l}(\rho)\,\vec{u}_\perp\|$, where $\rho = \|x_n\vec{v}\|$ is the length of the geodesic and

$$C_{\kappa_l}(s) = \begin{cases} \cos\left(\sqrt{\kappa_l}s\right) & ,\kappa_l > 0, \\ 1 & ,\kappa_l = 0, \\ \cosh\left(\sqrt{-\kappa_l}s\right) & ,\kappa_l < 0. \end{cases} \tag{15}$$

Therefore, we get

$$\sup_{\|\vec{u}\|=1} \|J_{\vec{u}}^{(x_n)}(1)\| \leq \sup_{\|\vec{u}\|=1} \sqrt{\|\vec{u}_{||}\|^2 + |C_{\kappa_l}(\rho)|^2\|\vec{u}_\perp\|^2} \tag{16}$$

$$\leq \max(1, |C_{\kappa_l}(\rho)|). \tag{17}$$

In $C_L(s)$, the maximum of $\cos$ is 1 so for non negative $\kappa_l$ our maximum will be 1. Additionally with $x_i \leq 1$, for the $\kappa_l < 0$ case we get $0 \leq \rho \leq 2(\tau_m + \tau)$ (see appendix C for detailed steps) and since $\cosh$ is monotonically increasing, $\cosh\left(\sqrt{-\kappa_l}\rho\right) \leq \cosh\left(2\sqrt{-\kappa_l}(\tau_m + \tau)\right)$, therefore

$$\sup_{\|\vec{u}\|=1} \|J_{\vec{u}}^{(x_n)}(1)\| \leq \begin{cases} 1 & ,\kappa_l \geq 0 \\ \cosh\left(2\sqrt{-\kappa_l}(\tau_m + \tau)\right) & ,\kappa_l < 0. \end{cases} \tag{18}$$

As this is independent of $x_n$, we get the sensitivity as,

$$\Delta_p \leq \begin{cases} \dfrac{2\tau}{n} & ,\kappa_l \geq 0, \\ \dfrac{2\tau}{n}\cosh(2\sqrt{-\kappa_l}(\tau_m + \tau)) & ,\kappa_l < 0. \end{cases} \tag{19}$$

$\square$

One can similarly get a bound on the sensitivity $\Delta_{\vec{v}}$ given by

$$\Delta_{\vec{v}} \leq \begin{cases} \dfrac{2\tau}{n} & ,\kappa_l \geq 0, \\ \dfrac{\tau}{n}\dfrac{\sinh(2\sqrt{-\kappa_l}(\tau_m + \tau))}{\sqrt{-\kappa_l}(\tau_m + \tau)} & ,\kappa_l < 0. \end{cases} \tag{20}$$

Refer to appendix D for the proof.

## 4 EXPERIMENTAL RESULTS

All experimental results were conducted on a 16 GB RAM laptop with Jupyter notebook on Windows 11. We denote the differentially private and non-private pair of footpoint and shooting vector as $(\tilde{p}, \tilde{v})$ and $(\hat{p}, \hat{v})$ respectively. The non-private estimates $(\tilde{p}, \tilde{v})$ that minimise the energy $E(p, v)$ are found using the geodesic regression code from Regmi (2020). We aim to measure how the private estimates affect the energy $E(p, v; D)$. To sample from the density, we utilize Riemannian Metropolis-Hastings, the Riemannian analog of the standard MCMC algorithm. We elaborate in appendix A specifically for the sphere and refer to optimization literature such as Absil et al. (2008) for more context.

We first sample a set of DP footpoints $(\tilde{p})$, each with privacy budget $\epsilon_p$, denoted as $\mathcal{P} = \{\tilde{p}_1, \tilde{p}_2, \ldots, \tilde{p}_m\}$. On the tangent space of *each* footpoint $\tilde{p}_i \in \mathcal{P}$, we sample a corresponding set of private shooting vectors $\mathcal{V}_i = \{\tilde{v}_{i1}, \tilde{v}_{i2}, \ldots, \tilde{v}_{im}\}$, each $\tilde{v}$ sampled at a privacy budget of $\epsilon_v$. Note that for each of the $m$ many footpoints we sample $m$ many shooting vectors on the respective tangent spaces yielding $m^2$ many private pairs.

We form a differentially private geodesic curve with a footpoint $\tilde{p}_i$ from the set $\mathcal{P}$ and a shooting vector $\tilde{v}_{ij}$ from the corresponding set $\mathcal{V}_i$. Each of these geodesics has a total privacy budget of $\epsilon = \epsilon_p + \epsilon_v$, by composition. Let $E_{ij} = \frac{1}{n}\sum_{k=1}^n d(\text{Exp}(p_i, x_k v_{ij}), y_k)^2$ be the Mean Squared Error (MSE) between the data points $\{y_k\}$ and the private predictions $\{\tilde{y}_k\}$ from the DP geodesic formed by the selected $\tilde{p}_i$ and $\tilde{v}_{ij}$. Next, let $\mathcal{E}_i = \{E_{i1}, E_{i2}, \ldots E_{im}\}$ be the set of MSE's with a fixed footpoint $p_i$ as the second index runs over the shooting vectors. Then $\bar{\mathcal{E}}_i$ is the average of such a set.

To asses our method, we consider the overall average MSE $\bar{\mathcal{E}} = \frac{1}{m} \sum_{i=1}^{m} \bar{\mathcal{E}}_i$ which aggregates the contributions from all sampled footpoints. We then analyze how this error changes under both equal and unequal allocations of the privacy budgets $(\epsilon_p, \epsilon_v)$.

Here we present results for the sphere and Kendall shape space. A similar analysis for the space of symmetric positive definite, $\mathrm{SPD}(2)$, matrices can be found in appendix F.

## 4.1 SPHERE

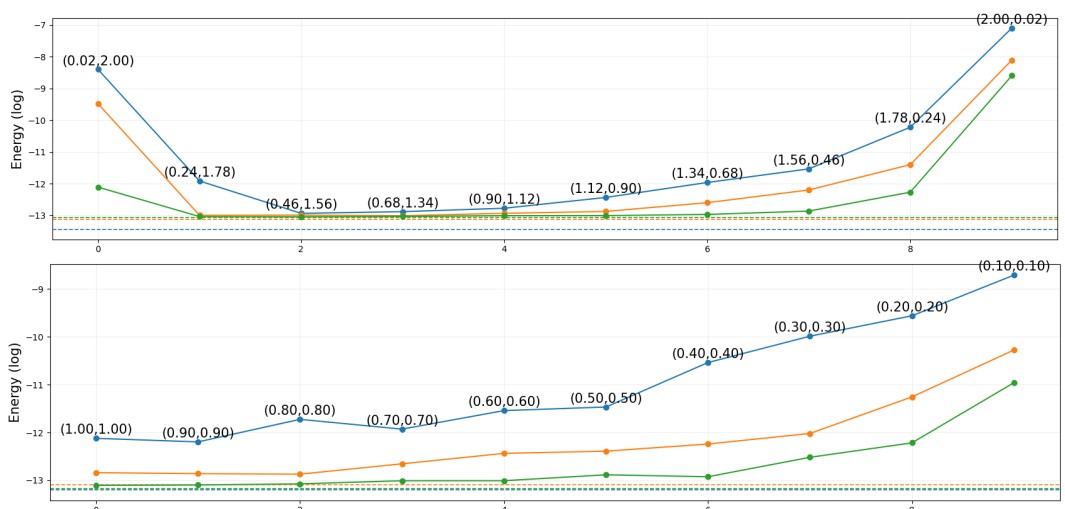

Figure 1: Log average MSE, $\ln \bar{\mathcal{E}}$, for $20$ (blue), $50$ (orange), and $100$ (green) data points on the sphere. Dotted line are the energies without privatisation. Top: unequal budget splits $\epsilon_p \in [0.02, 2.0]$, $\epsilon_v \in [2.0, 0.02]$, total $\epsilon = 2.02$ and Bottom: Equal budget splits with varying total budget $\epsilon \in [0.2, 2.0]$.

We create an artificial dataset of $20, 50, 100$ datapoints on a unit sphere $S_2$ by generating points on a geodesic and adding randomised errors to each point. That is, we parametrise a geodesic by randomly picking a point $q_0 \in S_2$ and a vector $\zeta \in T_{q_0}\mathcal{M}$. We then sample the geodesic $\gamma(t) = \mathrm{Exp}(q_0, t\zeta)$ uniformly on $t \in [0, 1]$, denote these as $\{\gamma_i\}_{i=1,\dots n}$. These $n$ points lie exactly on the geodesic. We add noise to the data points by sampling from a multivariate normal distribution with zero mean and a small covariance matrix ($\delta \times I_{3\times3}$), ensuring independent perturbations in each dimension. For this set of experiments, we use $\delta = 0.001$ and in appendix E we show results for $\delta = 0.01, 0.1$. Adding this noise, results in the data points lying outside the surface of the sphere therefore, we project the data points onto the unit sphere by normalization, i.e., $x \mapsto x/\|x\|$. We note that $q_0$ and $\zeta$ need not be parameters of geodesic regression due to the randomness of the injected noise.

For the unit sphere, the sensitivities of the footpoint and shooting vector are given by

$$\Delta_p = \frac{2\tau}{n}, \quad \Delta_v = \frac{2\tau}{n}.$$

Figure 1 illustrates how the average log mean squared error, $\ln \bar{\mathcal{E}}$, varies with $20, 50,$ and $100$ data points, shown by the blue, orange, and green lines, respectively. The dashed lines in each panel correspond to the geodesic log energy obtained from the non-private regression estimates $(\hat{p}, \hat{v})$.

For each privacy budget pair $(\epsilon_p, \epsilon_v)$, we sample $10$ candidate footpoints and $10$ shooting vectors per footpoint, resulting in $100$ private parameter pairs $(\tilde{p}, \tilde{v})$. The top panel corresponds to an unequal budget allocation, with $\epsilon_p \in [0.02, 2.0]$ and $\epsilon_v \in [2.0, 0.02]$ while maintaining a fixed total budget of $\epsilon = 2.02$. As expected, the log error is largest when either $\epsilon_p$ or $\epsilon_v$ is extremely small, producing a parabolic trend. A qualitatively similar pattern appears in the bottom panel, which depicts the case of equal budget allocation, where the total budget $\epsilon$ decreases from $2.0$ to $0.2$.

The increase in error at lower budgets is natural as with limited privacy allowance, the sampling distributions of $\tilde{p}$ and $\tilde{v}$ become heavy-tailed, leading to accepted samples that deviate substantially

from the non-private estimates $(\hat{p}, \hat{v})$, thereby inflating the energy. While the private energies approach the geodesic energy of the non-private estimates, they never exactly attain it. As the number of data points increases, the sensitivity decreases, and the sampling distribution of $(\tilde{p}, \tilde{v})$ becomes more concentrated around the non-private estimates. This results in energies that are increasingly close to the non-private geodesic energy, as seen in Figure 1. For each budget pair $(\epsilon_p, \epsilon_v)$, the log energy with 20 data points (blue) is higher than with 50 data points (orange), which in turn is higher than with 100 data points (green).

## 4.2 KENDALL SHAPE SPACE

Next, we analyze corpus callosum shapes from the Alzheimer's Disease Neuroimaging Initiative (ADNI) dataset processed by Cornea et al. (2017). We consider Alzheimer's patients aged 50–90 years, each represented by 50 uniformly sampled boundary landmarks. Landmark configurations are first mapped to the preshape sphere by removing translation (centering at the centroid) and scale (normalizing to unit Frobenius norm), so that all configurations satisfy zero mean and unit size. Kendall's shape space is formally the quotient of this preshape sphere by the action of rotations, yielding the complex projective space $\mathbb{C}P^{k-2}$. Following Fletcher's framework, however, we never explicitly project onto $\mathbb{C}P^{k-2}$; instead, we work directly in preshape coordinates, where the exponential and logarithm maps are expressed in forms that inherently respect rotational invariance. We also perform an affine transformation on the covariates (ages) such that they are scaled to be between $[0, 1]$. The sectional curvature of Kendall shape space with landmarks $\geq 4$ is bounded

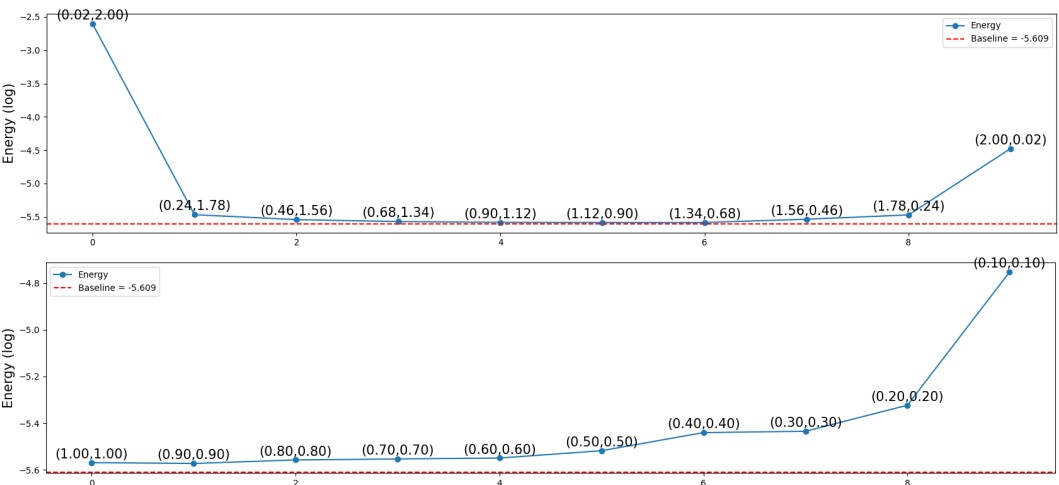

Figure 2: Log average MSE, $\ln \bar{\mathcal{E}}$, for 100 data points on $\mathbb{C}P^{k-2}$. Dotted line is the log energy without privatisation. Top: unequal budget splits $\epsilon_p \in [0.02, 2.0]$, $\epsilon_v \in [2.0, 0.02]$, total $\epsilon = 2.02$ and Bottom: Equal budget splits with varying total budget $\epsilon \in [0.2, 2.0]$.

between $[1, 4]$ resulting in $\kappa_l = 1$. We get the sensitivities of the footpoint and shooting vector as,

$$\Delta_p = \frac{2\tau}{n}, \quad \Delta_v = \frac{2\tau}{n}.$$

For Kendall's shape space, we examine the log average MSE under both unequal and equal budget allocations, as shown in Figure 2. Similar to the case of unit sphere, for each privacy budget pair $(\epsilon_p, \epsilon_v)$, we sample 10 candidate footpoints and 10 shooting vectors per footpoint, resulting in 100 private parameter pairs $(\tilde{p}, \tilde{v})$ The overall behavior closely parallels the trends observed on the unit sphere. In the case of unequal allocation, when either $\epsilon_p$ or $\epsilon_v$ is extremely small, the log error rises sharply, producing a parabolic profile. This is explained by the heavy-tailed sampling distribution at low budgets, which increases the likelihood of accepting footpoints and shooting vectors far from the true regression parameters. Under a more balanced (equal) split of the budget, the errors decrease as the total privacy budget increases, reflecting a concentration of the sampling distribution around the regression estimates. Together, these results demonstrate that while Kendall's shape space is geometrically more complex than the unit sphere, the qualitative relationship between privacy

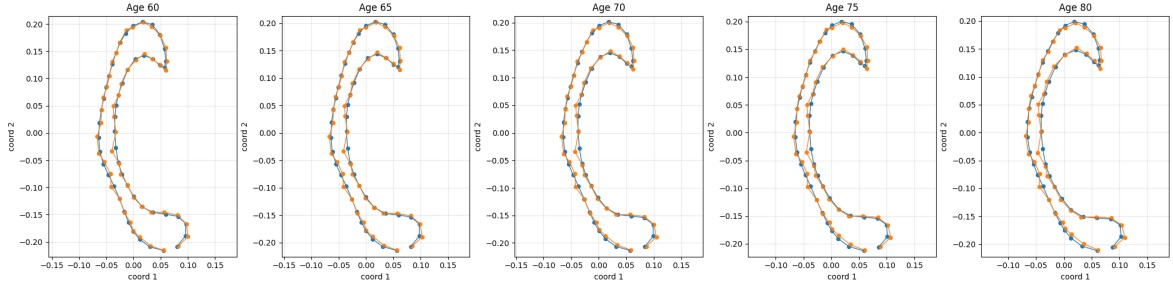

Figure 3: Corpus callosum shapes at ages 60, 65, 70, 75, and 80. Blue curves denote predictions from the non-private regression parameters, while orange curves correspond to predictions from the differentially private parameters with $\epsilon_p = \epsilon_v = 0.3$.

allocation and estimation accuracy remains consistent: extreme imbalance in budget allocation leads to instability, whereas balanced splits yield more reliable regression fits. Figure 3 provides a visual comparison of predictions obtained from non-private and differentially private regression parameters under a budget split of $\epsilon_p = \epsilon_v = 0.3$. For this choice of privacy allocation, the predicted shapes from the private parameters align closely with those from the non-private regression, with only minor deviations visible across ages.

## 5 DISCUSSION

Geodesic regression is an extension of linear regression to Riemannian manifolds. We have developed a methodology to release the parameters of geodesic regression (the footpoint and shooting vector) in a differentially private manner. To do so, we derive a theoretical bound on the sensitivity, under the KNG mechanism, of each parameter. We note that the theoretical bounds on the sensitivity requires Riemannian manifolds with curvature bounded from above. This, however, is not a limitation specific to DP as it is also necessary for the Fréchet mean and for the parameters of geodesic regression. We show that the sensitivity of the parameters is a function of its respective Jacobi fields which itself is a function of the curvature of the manifold. A similar discovery was previously noted for the estimating the Fréchet mean by Reimherr et al. (2021). We thus demonstrate our methodology on the 2-sphere, Kendall's shape space, and the space of symmetric positive definite matrices over a variety of sample sizes, variety of budgets, and unequal allocation of budgets in sections 4, E, and F.

Similar to the standard exponential mechanism in Euclidean space, it is not straightforward to sample from KNG. This difficulty is compounded by the difficulty of sampling on manifolds. We rely on MCMC which has its limitations for privacy (Wang et al., 2015); however, developing sampling algorithms is itself a research area outside the scope of this paper. Another problem one typically encounters is bounding the data. A usual assumption is $D \subset B_r(m)$, a ball of radius $r$ centered at $m$. For most data one cannot know $r$ a priori. We face a similar issue as we need $\tau = \sup \|\varepsilon\|$ which we cannot know beforehand. We set $\tau = \max_i \|\varepsilon_i\|$ which indeed violates privacy as we need to look at the data, but the concept of our methodology still holds. Further, since $\sigma = 2\tau/n\epsilon$ and we experiment under different $\epsilon$ we, in a sense, verify our methodology holds if we had inflated $\tau$.

There are many avenues for future research. For instance, studying how DP interacts in the framework of multiple geodesic regression, Fréchet regression, or Wasserstein regression are all significant contributions. Further interestingly, parametrising a geodesic poses an opportunity unique to manifolds. For instance, one can reverse the parametrisation of a geodesic $\gamma : [0, 1] \to \mathcal{M}$ to $\tilde{\gamma}(t) = \gamma(1 - t)$ One might consider such an endeavor if, for instance, the curvature of the manifold is *different* at the endpoints, i.e. $\gamma(0)$ and $\gamma(1)$, thus directly affecting the sensitivity of $\tilde{p}$. Our methodology relies on an upper bound on the curvature but if more information is known, one can possibly leverage this information. Another possible is utilizing DP Riemannian optimization Han et al. (2024a); Utpala et al. (2023). This route may alleviate the problem utilizing an approximate sampler but has its accompanying methodological difficulties.

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

# A  SAMPLING

This section describes how one could sample from the KNG mechanism on the unit sphere. We implement a Riemannian random walk Metropolis-Hastings algorithm to sample from the KNG mechanism. Suppose we have bounds on the sensitivity of $p$ and $v$, denoted $\Delta_p$ and $\Delta_v$ respectively, as in 3.

We have that $f(p; D) \propto \exp\{-\sigma^{-1}\|\nabla_p E(p; D)\|_p\}$. Given that $\sigma = \Delta_p/\epsilon_p$ and the gradient is as in 3,

$$
\begin{aligned}
f(p; D) &\propto \exp\left\{-\frac{\|\nabla_p E(p; D)\|_p}{\Delta_p/\epsilon_p}\right\} \\
&= \exp\left\{-\frac{\|\nabla_p 1/n \sum d^2(\text{Exp}(p, x_i v), y_i)\|_p}{2\tau/(n\epsilon_p)}\right\} \\
&= \exp\left\{-\frac{\|\sum d_p\text{Exp}(p, x_i v)^\dagger \varepsilon_i\|_p}{2\tau/\epsilon_p}\right\}.
\end{aligned}
$$

We note that for a negatively curved manifold, another term is required in the denominator. To sample from this distribution, first, initialize the random walk at $p_0$ which can be any data point, the Fréchet mean of the points, or the sample statistic $\hat{p}$. To make a proposal we first sample a vector $\nu_{prop}$ from the tangent space, as it is simply a Euclidean plane, uniformly around $p_0$ with a radius $\eta$. This $\eta$ is a tuning parameter that determines the 'stickiness' of the MCMC chain. We set $\eta \propto \sigma$. The proposal is then $p_{prop} := \text{Exp}(p_0, \nu_{prop})$. Typically one would then compute the acceptance probability, however, note that the energy depends on both the shooting vector *and* a footpoint. Since we are under the assumption $v$ is constant, we parallel transport $v$ to the proposal that is, we use $\Gamma_{\hat{p}}^{p_{prop}}\hat{v}$. Lastly, compute the acceptance probability in the typical fashion $f(p_{prop}; D)/f(p_0; D)$. We draw subsequent instances in the same fashion continuing the random walk from the previously accepted footpoint. We forego thinning the chain as it has been argued that it is not a necessary procedure (Link and Eaton, 2012). Furthermore, thinning decreases efficiency which compounds with the inefficiency of sampling on manifolds.

While sampling instead of $\tau$ we use $\max \|\varepsilon_i\|$, the empirical largest error, in the acceptance probability as the largest error in the dataset. This is not ideal as this value is data driven and thus is not private. This is unfortunately a problem often faced in DP. For instance, when sanitizing the mean of real numbers a common assumptions is that the data live in a bounded ball $B_r(y)$ centered at $y$ with radius $r > 0$. This ball can be determined by utilizing public information. In our setting we foresee a similar solution. For example, if one were to use geodesic regression on Earth to model the migration of birds, one could use the guidance of bird experts to select the maximum deviation.

## A.1  PRIVACY AND MCMC

To sample from KNG on Riemannian manifolds we utilize the Metropolis-Hastings MCMC algorithm. There has been a lot of literature on how MCMC impacts the privacy implications, see for instance Bertazzi et al. (2025); Seeman et al. (2021). Roughly speaking, a pure DP mechanism, $(\epsilon, \delta = 0)$, which utilizes an approximate sampler such as Metropolis-Hastings, will satisfy approximate-DP with $\delta = O(1/M)$ where $M$ is the length of chain, under some assumptions of proper mixing. This does then weaken the privacy guarantees of our simulations. However, we note two important considerations. First, for Riemannian manifolds samplers are not as widely available as for Euclidean spaces. For Riemannian manifolds, samplers need to be developed for each manifold with its respective metric. For instance, the Laplace on the space of symmetric positive-definite matrices with the affine metric is studied in Hajri et al. (2016), but if one were to use the same space but change the metric to Bures-Wasserstein or Log-Euclidean, this would require a different set up. Even in Euclidean space, exact samplers for a general exponential mechanism are not guaranteed. Second, we note that in practice we would only release one private pair $(\tilde{p}, \tilde{v})$, so being cognizant of that $\delta = O(1/M)$ one can control the impact on the privacy implications. That is, one can set $M$ arbitrarily large to arrive at a negligible $\delta$.

# B  RIEMANNIAN GEOMETRY

Following Nakahara (2018) we elaborate on two important aspects of Riemannian geometry that are widely used in this paper.

## B.1  PARALLEL TRANSPORT

On a Riemannian manifold, the metric $g$ induces a natural volume form with the help of a local basis $\{\partial/\partial x^i\}$ given by $\text{vol}_g = \sqrt{\det_g}\,dx^1 \wedge dx^2 \cdots \wedge dx^n$. The Riemannian measure $d\mu_g$ is derived using the volume form and allows us to integrate over the manifold,

$$d\mu_g = \sqrt{\det_g}\,dx^1 dx^2 \cdots dx^n.$$

Any subset of the manifold $A \subset \mathcal{M}$ is measurable if it belongs to the $\sigma$- algebra $\mathcal{M}$ associated with the Riemmanian measure.

The generalization of a Euclidean shortest path, straight lines, on Riemannian manifolds is referred to as a "geodesic." A connection $\nabla$ on a manifold can be used to take directional derivatives and thus define a geodesic curve. A curve $\gamma : [0,1] \to \mathcal{M}$ on $\mathcal{M}$ is a geodesic curve with respect to $\nabla$ if its acceleration is zero i.e. $\nabla \dot\gamma = 0$. On a manifold with linear connection there always exists a unique geodesic which is denoted by a footpoint $p$ and shooting vector $v \in T_p\mathcal{M}$. The distance, $d(\cdot,\cdot)$, between two points $p, q \in \mathcal{M}$ is thus the length of the geodesic between them $d(p,q) := \mathcal{L}(\gamma) = \int \|\dot\gamma(t)\|_{\gamma(t)}^{1/2}\mathrm{d}t$.

Parallel transport plays a crucial role in proving theorem D.1 and sampling replicates for the MCMC chain. Unlike Euclidean space comparing two vectors on a general manifold $\mathcal{M}$ becomes more challenging as the vector can lie on different tangent spaces of $\mathcal{M}$. Consider two points on the manifold close to each other, $x, x + \delta x$. We can have a vector field on the tangent space of $x$ given by $V = V^\mu e_\mu$ where $e_\mu$ is the local basis and $V_\mu$ are the vector components. In Euclidean space the derivative with respect to $x^\nu$ is given by:

$$\frac{\partial V\mu}{\partial x^\nu} = \lim_{\delta x \to 0} \frac{V^\mu(\cdots, x^\nu + \delta x^\nu, \cdots) - V^\mu(\cdots, x^\nu, \cdots)}{\delta x^\nu} \tag{21}$$

On a general manifold, we need to transport $V^\mu(x)$ to $x + \delta x$ to perform the above subtraction. Denote a vector $V(x)$ transported to $x + \delta x$ as $\tilde{V}(x + \delta x)$ and satisfies the following conditions,

$$\tilde{V}^\mu(x + \delta x) - V^\mu(x) \propto \delta x \tag{22}$$

$$\widetilde{V^\mu + W^\mu} = \tilde{V}^\mu(x + \delta x) + W^\mu(x + \delta x). \tag{23}$$

Where $W = W^\mu e_\mu$ is another vector field on $x$. A transport is called *parallel transport* if the above conditions are satisfied. If we take $\tilde{V}^\mu(x + \delta x) = V^\mu(x) - V^\lambda(x)\Gamma_{\nu\lambda}^\mu \delta x^\nu$, the above rules are satisfied.. There are distinct rules of parallel transport and each one is written with a choice of connection, $\Gamma$. For a manifold with a metric, there is a preferred choice of $\Gamma$ called as Levi-Civita connection to define the parallel transport. Using the connection we can thus define a covariant derivative which is similar to a directional derivative in Euclidean space as,

$$\nabla_\mu V^\lambda = \frac{\partial W^\lambda}{\partial x^\mu} + \Gamma_{\mu\nu}^\lambda W^\nu, \tag{24}$$

where $\nabla_\mu W^\lambda$ is the $\lambda$th component of a vector $\nabla_\mu W$.

An important theorem in Riemannian geometry is the Hopf-Rinow theorem which states that:

**Theorem B.1.** *Let $(\mathcal{M}, g)$ be a connected finite-dimensional Riemannian manifold and let $d$ be the distance induced by $g$. The following are equivalent:*

1. *$(M, d)$ is a complete metric space.*

2. *Every closed and bounded subset of $\mathcal{M}$ is compact.*

3. *$\mathcal{M}$ is geodesically complete, that is, every geodesic $\gamma : (a, b) \to \mathcal{M}$ can be extended to a geodesic defined on all of $\mathbb{R}$.*

*Moreover, if these conditions hold, then for any $p, q \in \mathcal{M}$ there exists a minimizing geodesic segment $\gamma : [0, 1] \to \mathcal{M}$ such that*

$$\gamma(0) = p, \qquad \gamma(1) = q, \qquad d(p, q) = \text{length}(\gamma).$$

The fact that a minimizing geodesic between two points in a strongly convex ball is unique ensures that parallel transport is unique. This is given by the following theorem.

**Theorem B.2.** *Suppose $p, q \in B_r(m_0)$. Then there exists a unique minimizing geodesic $\gamma : [0, 1] \to \mathcal{M}$ that joins $p$ to $q$. Parallel transport along $\gamma$ with respect to the Levi–Civita connection therefore defines a unique linear isometry*

$$\Gamma_q^p : T_q\mathcal{M} \longrightarrow T_p\mathcal{M}.$$

*In particular, for any $v \in T_p\mathcal{M}$ and $w \in T_q\mathcal{M}$, the quantity*

$$\left\| v - \Gamma_q^p(w) \right\|$$

*is well defined, depends only on the pair $(p, q)$, and is independent of any choice of paths.*

In our work, we assume that the data lies in a geodesically convex ball and therefore, transporting vectors from $T_q\mathcal{M}$ to $T_p\mathcal{M}$ along the unique minimizing geodesic $\gamma$ yields a uniquely determined linear map $\Gamma_q^p : T_q\mathcal{M} \to T_p\mathcal{M}$. Since the Levi–Civita connection is metric compatible and torsion free, parallel transport preserves the Riemannian inner product and hence the induced norm, so $\Gamma_q^p$ is an isometry. It follows that for any $v \in T_p\mathcal{M}$ and $w \in T_q\mathcal{M}$, the difference $v - \Gamma_q^p(w)$ is unambiguously defined in $T_p\mathcal{M}$, and its norm $\|v - \Gamma_q^p(w)\|$ does not depend on any choice of path other than the minimizing geodesic.

### B.2 RIEMANNIAN CURVATURE TENSOR

The geometric meaning of the curvature of a manifold and the Riemann curvature tensor is understood by parallel transporting a vector $V_0$ at $p$ to a different point $q$ along two distinct curves $C$ and $C'$. One can notice in Figure 4 that the resulting two vectors are different from each other. This non-integrability of parallel transport defines the intrinsic notion of curvature of a manifold.

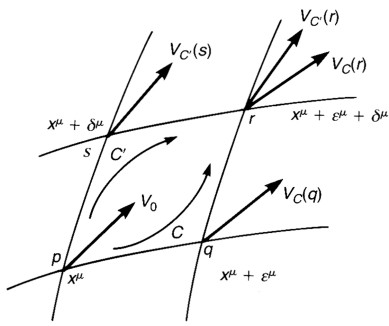

Figure 4: The vector $V_0$ is parallel transported along the curves $C$ and $C'$ resulting in $V_C(r)$ and $V_{C'}(r)$. The difference between these resulting vectors at the point $r$ represents the curvature of the manifold (Nakahara, 2018).

Take four points on a manifold defined by the vertices of an infinitesimal parallelogram, $p \equiv x^\mu, q \equiv x^\mu + \epsilon^\mu, s \equiv x^\mu + \delta^\mu, r \equiv x^\mu + \delta^\mu + \epsilon^\mu$. We can parallel transport a vector $V_0$ along two curves defined by $C = p\,q\,r$ and $C' = p\,s\,r$. The resulting vectors $V_C^\mu(r)$ and $V_{C'}$ can be written in terms of the original vector $V_0 \in T_p\mathcal{M}$ as,

$$V_C^\mu(r) = V_0^\mu - V_0^\kappa \Gamma_{\nu\kappa}^\mu(p)\epsilon^\nu - V_0^\kappa \Gamma_{\nu\kappa}^\mu(p)\delta^\nu - V_0^\kappa(\partial_\lambda \Gamma_{\nu\kappa(p)}^\mu - \Gamma_{\lambda\kappa}^\rho(p)\Gamma_{\nu\rho}(p))\epsilon^\lambda \delta^\nu \qquad (25)$$

$$V_{C'}^\mu(r) = V_0^\mu - V_0^\kappa \Gamma_{\nu\kappa}^\mu(p)\delta^\nu - V_0^\kappa \Gamma_{\nu\kappa}^\mu(p)\epsilon^\nu - V_0^\kappa(\partial_\nu \Gamma_{\lambda\kappa(p)}^\mu - \Gamma_{\nu\kappa}^\mu(p)\Gamma_{\lambda\rho}(p))\epsilon^\lambda \delta^\nu. \qquad (26)$$

Once we have parallel transported the vectors on the tangent space of $r$ we can quantify their difference as

$$V_{C'}(r) - V_C(r) = V_0^\kappa(\partial_\lambda \Gamma_{\nu\kappa}^\mu(p) - \partial_\nu \Gamma_{\nu\kappa}^\mu(p) - \Gamma_{\lambda\kappa}^\rho(p)\Gamma_{\nu\rho}^\mu(p) + \Gamma_{\nu\kappa}^\rho(p)\Gamma_{\lambda\rho}^\rho(p))\epsilon^\lambda \delta^\nu \qquad (27)$$

$$= V_0^\kappa R_{\kappa\lambda\nu}^\mu \epsilon^\lambda \delta^\nu. \qquad (28)$$

The Riemann curvature tensor ($R_{\kappa\lambda\nu}^\mu$) is defined as this difference and represents the curvature.

### B.3 Proof for Gradient of Energies

We will prove

$$\nabla_p E(p, v) \;=\; -\frac{1}{n}\sum_{i=1}^{n}\big(d_p\mathrm{Exp}(p, x_i v)\big)^\dagger \varepsilon_i, \qquad \varepsilon_i := \mathrm{Log}(\eta_i, y_i), \;\; \eta_i := \mathrm{Exp}(p, x_i v), \quad (29)$$

where $\dagger$ denotes the adjoint operator. For a linear map $A : T_p\mathcal{M} \to T_\eta\mathcal{M}$ we define $A^\dagger : T_\eta M \to T_p\mathcal{M}$ by

$$\langle Au, w\rangle_\eta \;=\; \langle u, A^\dagger w\rangle_p \quad \text{for all } u \in T_p\mathcal{M}, \; w \in T_\eta M.$$

Given the smooth function $F(\eta) = \frac{1}{2}d^2(\eta, y)$, one can easily prove that $\nabla_\eta F(\eta) = -\mathrm{Log}(\eta, y)$. We will also use the fact that the Riemannian gradient is that which is uniquely defined in terms of the directional derivative as:

$$\frac{d}{ds}\bigg|_{s=0} F(\eta(s)) \;=\; dF(\eta)[w] \;=\; \langle \nabla F(\eta), w\rangle. \quad (30)$$

This follows from differentiating a function $F : \mathcal{M} \to \mathbb{R}$ along a curve $\eta(s)$ with $\dot\eta(0) = w$.

Next, for fixed $p \in \mathcal{M}$ and $v \in T_p\mathcal{M}$, consider the geodesic $\gamma(t) = \mathrm{Exp}(p, tv)$. Let $p_s := p(s)$ be a smooth curve with $p_0 = p$ and $\dot p_0 = u \in T_p\mathcal{M}$. Further, let $v_s \in T_{p_s}\mathcal{M}$ be the parallel transport of $v$ along $p_s$ (so $\nabla_s v_s|_{s=0} = 0$). Define the variation by geodesics

$$c(s, t) \;=\; \mathrm{Exp}\big(p_s, \, t\, v_s\big).$$

Then $J(t) := \partial_s c(0, t)$ is a Jacobi field along $\gamma$ with initial conditions $J(0) = u$, $J'(0) = 0$ and

$$d_p\mathrm{Exp}(p, v)[u] \;=\; \partial_s c(0, 1) \;=\; J(1). \quad (31)$$

We want the change in geodesic energy if we perturb $p$ along $p_s$. Using $\eta_i(s) = \mathrm{Exp}(p_s, x_i v)$ together with the chain rule and equation 30 we obtain,

$$\nabla_p E = \frac{1}{2n}\frac{d}{ds}\bigg|_{s=0} d\big(\eta_i(s), y_i\big)^2 = \frac{1}{2n}\big\langle \nabla_\eta d(\eta, y_i)^2\big|_{\eta=\eta_i(0)}, \dot\eta_i(0)\big\rangle = \frac{1}{n}\big\langle -\varepsilon_i, \dot\eta_i(0)\big\rangle_{\eta_i}$$

where $\dot\eta_i(0) = d_p\mathrm{Exp}(p, x_i v)[u]$. Hence

$$\frac{d}{ds}\bigg|_{s=0}\frac{1}{2}d\big(\eta_i(s), y_i\big)^2 = \big\langle -\varepsilon_i, d_p\mathrm{Exp}(p, x_i v)[u]\big\rangle_{\eta_i}.$$

Summing over $i$ we have,

$$\frac{d}{ds}\bigg|_{s=0} E(p_s, v) = \frac{1}{n}\sum_{i=1}^{n}\big\langle -\varepsilon_i, d_p\mathrm{Exp}(p, x_i v)[u]\big\rangle = \frac{1}{n}\Big\langle u, \sum_{i=1}^{n}\big(d_p\mathrm{Exp}(p, x_i v)\big)^\dagger(-\varepsilon_i)\Big\rangle_p,$$

where the last equality is the definition of the adjoint. Because this holds for every $u \in T_p\mathcal{M}$, the Riemannian gradient at $p$ is

$$\nabla_p E(p, v) \;=\; -\frac{1}{n}\sum_{i=1}^{n}\big(d_p\mathrm{Exp}(p, x_i v)\big)^\dagger \varepsilon_i, \quad (32)$$

as claimed. One can follow similar steps for the gradient with respect to $v$ to derive,

$$\nabla_v E(p, v) \;=\; -\frac{1}{n}\sum_{i=1}^{n} x_i\big(d_v\mathrm{Exp}(p, x_i v)\big)^\dagger \varepsilon_i. \quad (33)$$

## C  Bound on Geodesic Length $\rho$

Let $\mathcal{M}$ be a Riemmanian manifold with sectional curvature bounded as $\kappa_l \le \kappa \le \kappa_h$ and $\kappa_l < 0$. $\{(x_i, y_i)\}_{i=1}^{N}$ be data with $y_i \in \mathcal{M}$ and suppose there exists $m_0 \in \mathcal{M}$ and $\tau_m > 0$ such that

$$y_i \in B_{\tau_m}(m_0) \qquad \text{for all } i.$$

Before performing the regression task we reparameterise the predictors so that for some index $k$ we have $x_k = 0$. Consider the geodesic regression model

$$\hat{y}_i = \text{Exp}(\hat{p}, x_i \hat{v}),$$

and assume the fit is $\tau$-close:

$$d(\hat{y}_i, y_i) \leq \tau \qquad \text{for all } i.$$

Define the geodesic reach from the intercept

$$\rho := \max_i d(\hat{p}, \hat{y}_i).$$

Since $x_k = 0$, we have $\hat{y}_k = \text{Exp}(\hat{p}, 0) = \hat{p}$. By $\tau$-closeness, $d(\hat{p}, y_k) = d(\hat{y}_k, y_k) \leq \tau$. Because $y_k \in B_{\tau_m}(m_0)$, we have $d(y_k, m_0) \leq \tau_m$. The triangle inequality gives

$$d(\hat{p}, m_0) \leq d(\hat{p}, y_k) + d(y_k, m_0) \leq \tau + \tau_m.$$

Now, fix any $i$. By the triangle inequality and the $\tau$-closeness assumption,

$$\begin{aligned} d(\hat{p}, \hat{y}_i) &\leq d(\hat{p}, y_i) + d(y_i, \hat{y}_i) \\ &\leq \big(d(\hat{p}, m_0) + d(m_0, y_i)\big) + \tau \\ &\leq (\tau + \tau_m) + \tau_m + \tau \\ &= 2(\tau_m + \tau). \end{aligned}$$

Taking the maximum over $i$ yields $\rho \leq 2(\tau_m + \tau)$.

## D    SENSITIVITY BOUND FOR THE SHOOTING VECTOR

In section 3 it was shown how to bound the sensitivity for the footpoint. In this section, we give the theorem with proof to bound the sensitivity for the shooting vector. As before we will use the KNG mechanism focusing on the shooting vector. If the assumption 3.1 and 3.2 are satisfied the bound on the sensitivity $\Delta_v$ is given by:

**Theorem D.1.** *Let Assumptions 3.1 and 3.2 hold. Let $D, D'$ be adjacent datasets, for a fixed shooting vector $v$,*

$$\begin{aligned} \Delta_{\vec{v}} &= \sup_{D \sim D'} \|\nabla_{\vec{v}} E(p; D) - \nabla_{\vec{v}} E(p; D')\|, \\ &\leq \frac{2\tau}{n}, \kappa_l \geq 0, \\ &\leq \frac{2\tau}{n} \frac{\sinh\left(\sqrt{-\kappa_l}\tau\right)}{\sqrt{-\kappa_l}}, \kappa_l < 0 \end{aligned}$$

*Proof.* We will start with fixing the footpoint $p$ and focus on the shooting vector, $\vec{v}$. The global sensitivity is

$$\Delta_{\vec{v}} = \sup_{D \sim D'} \|\nabla_{\vec{v}} E(p; D) - \nabla_{\vec{v}} E(p; D')\|. \tag{34}$$

Using the chain rule, the gradient of the energy is given by, $\nabla_{\vec{v}} E(p; D) = -\frac{1}{n} \sum_{i=1}^{n} x_i \, d_{\vec{v}} \text{Exp}(p, x_i v)^{\dagger} \vec{\varepsilon_i}$ where $\vec{\varepsilon_i} = \text{Log}(\text{Exp}(p, x_n v), \hat{y}_n)$ is the error vector and $\dagger$ denotes the adjoint operator. The norm in equation 34 is thus the difference of two sums that differ in only one term due to the adjacent datasets $D \sim D'$. All terms, thus, cancel except the last. We have,

$$\Delta_{\vec{v}} = \frac{1}{n} \big\| x_n \, d_{\vec{v}} \text{Exp}(p, x_n v)^{\dagger} \vec{\varepsilon_n} - x'_n \, d_{\vec{v}} \text{Exp}(p, x'_n v)^{\dagger} \vec{\varepsilon_n}' \big\|, \tag{35}$$

$$\leq \frac{1}{n} \Big( \|x_n \, d_{\vec{v}} \text{Exp}(p, x_n v)^{\dagger} \vec{\varepsilon_n}\| + x'_n \|d_{\vec{v}} \text{Exp}(p, x'_n v)^{\dagger} \vec{\varepsilon_n}'\| \Big), \tag{36}$$

$$\leq \frac{1}{n} \Big( |x_n| \|d_{\vec{v}} \text{Exp}(p, x_n v)^{\dagger}\|_{op} \|\vec{\varepsilon_n}\| + |x'_n| \|d_{\vec{v}} \text{Exp}(p, x'_n v)^{\dagger}\|_{op} \|\vec{\varepsilon_n}'\| \Big), \tag{37}$$

$$\leq \frac{1}{n} \Big( \|d_{\vec{v}} \text{Exp}(p, x_n v)\|_{op} \|\vec{\varepsilon_n}\| + \|d_{\vec{v}} \text{Exp}(p, x'_n v)\|_{op} \|\vec{\varepsilon_n}'\| \Big), \tag{38}$$

$$\leq \frac{\tau}{n} \Big( \sup_{\|\vec{u}\|=1} \|J_{\vec{u}}^{(x_n)}(1)\| + \sup_{\|\vec{u}\|=1} \|J_{\vec{u}}^{(x'_n)}(1)\| \Big) \tag{39}$$

Here the second step applies the triangle inequality, the third step uses the characterization $\|A\vec{\varepsilon_n}\| \leq \|A\|_{op}\|\vec{\varepsilon_n}\|$ for linear operators, and the operator norm is preserved under the adjoint, i.e. $\|A^\dagger\| = \|A\|$ and $x_n, x'_n \in [0,1]$ we can substitute $x_n = x'_n = 1$ to get the worst upper bound. In the final step we use the definition of operator norm and equation 8 to get $\|d_{\vec{v}}\text{Exp}(p, x_n v)\|_{op} = \sup_{\|\vec{u}\|=1} \|d_{\vec{v}}\text{Exp}(p, x_n v) \vec{u}\| = \sup_{\|\vec{u}\|=1} \|J_{\vec{u}}^{(x_n)}(1)\|$ where $\vec{u}$ is deviation of $\vec{v}$. The initial conditions on $J'^{(x_n)}_{\vec{u}}(t)$ are $J'^{(x_n)}_{\vec{u}}(0) = \vec{u}, J^{(x_n)}_{\vec{u}}(0) = 0$. The $\tau$ comes out as we have assumed that the data is $\tau$-close, with $\|\varepsilon_n\|, \|\varepsilon'_n\| < \tau$ in line with assumption 3.2.

Next let's consider only the first part of equation 39 (second part will follow the same analysis). We will use the Rauch comparison theorem by taking the model manifold with constant sectional curvature $\kappa = \kappa_l$.

$$\sup_{\|\vec{u}\|=1} \|J_{\vec{u}}^{(x_n)}(1)\| \leq \sup_{\|\vec{u}\|=1} \|\tilde{J}_{\vec{u}}^{(x_n)}(1)\|$$

Where $\tilde{J}$ is the Jacobi field on our model manifold $\tilde{\mathcal{M}}$. We can next decompose $\vec{u}$ in perpendicular and parallel components to the geodesic, $\|\tilde{J}_{||}(1)\| = \|\vec{u}_{||}\|$ and $\tilde{J}_\perp$ is dependent on the curvature, $\|\tilde{J}_\perp(1)\| = \|\frac{S_{\kappa_l}(\rho)}{\rho} \vec{u}_\perp\|$, where $\rho = |x_n \vec{v}|$ is the length of the geodesic.

$$S_{\kappa_l}(s) = \begin{cases} \frac{1}{\sqrt{\kappa_l}} \sin\left(\sqrt{\kappa_l}s\right) & , \kappa_l > 0, \\ s & , \kappa_l = 0, \\ \frac{1}{\sqrt{-\kappa_l}} \sinh\left(\sqrt{-\kappa_l}s\right) & , \kappa_l < 0 \end{cases} \tag{40}$$

Therefore, we get:

$$\sup_{\|\vec{u}\|=1} \|J_{\vec{u}}^{(x_n)}(1)\| \leq \sup_{\|\vec{u}\|=1} \sqrt{\|\vec{u}_{||}\|^2 + |\frac{S_{\kappa_l}(\rho)}{\rho}|^2 \|\vec{u}_\perp\|^2} \tag{41}$$

$$\leq \max(1, |\frac{S_{\kappa_l}(\rho)}{\rho}|) \tag{42}$$

In $S_{\kappa_l}(s)$ maximum of sin is 1 so for non negative $\kappa_l$ our maximum will be 1. For $\kappa_l < 0$, we have $x_n \in [0,1]$ and $0 \leq \rho \leq 2(\tau_m + \tau)$ (check appendix C for detailed steps) and since sinh is monotonically increasing $\sinh\left(\sqrt{-\kappa_l}\rho\right) \leq \sinh\left(2\sqrt{-\kappa_l}(\tau_m + \tau)\right)$, therefore :

$$\sup_{\|\vec{u}\|=1} \|J_{\vec{u}}^{(x_n)}(1)\| \leq \begin{cases} 1 & , \kappa_l \geq 0 \\ \frac{1}{2\sqrt{-\kappa_l}(\tau_m+\tau)} \sinh\left(2\sqrt{-\kappa_l}(\tau_m + \tau)\right) & , \kappa_l < 0. \end{cases} \tag{43}$$

As this is independent of $x_n$, finally we get the sensitivity as:

$$\Delta_{\vec{v}} \leq \begin{cases} \dfrac{2\tau}{n}, & \kappa_l \geq 0, \\ \dfrac{\tau}{n} \dfrac{\sinh(2\sqrt{-\kappa_l}(\tau_m + \tau))}{\sqrt{-\kappa_l}(\tau_m + \tau)}, & \kappa_l < 0. \end{cases} \tag{44}$$

$\square$

# E  EFFECT OF DATA SPREAD ON UNIT SPHERE

For the experiments on the unit sphere, recall that data points were generated by perturbing locations on the sphere with Gaussian noise sampled from a multivariate normal distribution with zero mean and covariance matrix $\delta I_{3\times3}$, fixing $\delta = 0.001$ in the main text. In this appendix, we study the effect of larger noise by increasing $\delta$ to 0.01 and 0.1. Figure 5 provides a visual illustration of the data together with the non-private geodesic and the private geodesics obtained under equal allocation of the privacy budget, with total budget $\epsilon \in [2.0, 0.2]$. The geodesics are color–coded according to their total privacy budgets: darker blue indicates a smaller total budget. For each row, the second panel provides a zoomed–in view for clarity. In the top row ($\delta = 0.01$), some private geodesics with smaller budgets (shown in purple and blue) deviate farther from the data. When the noise level increases

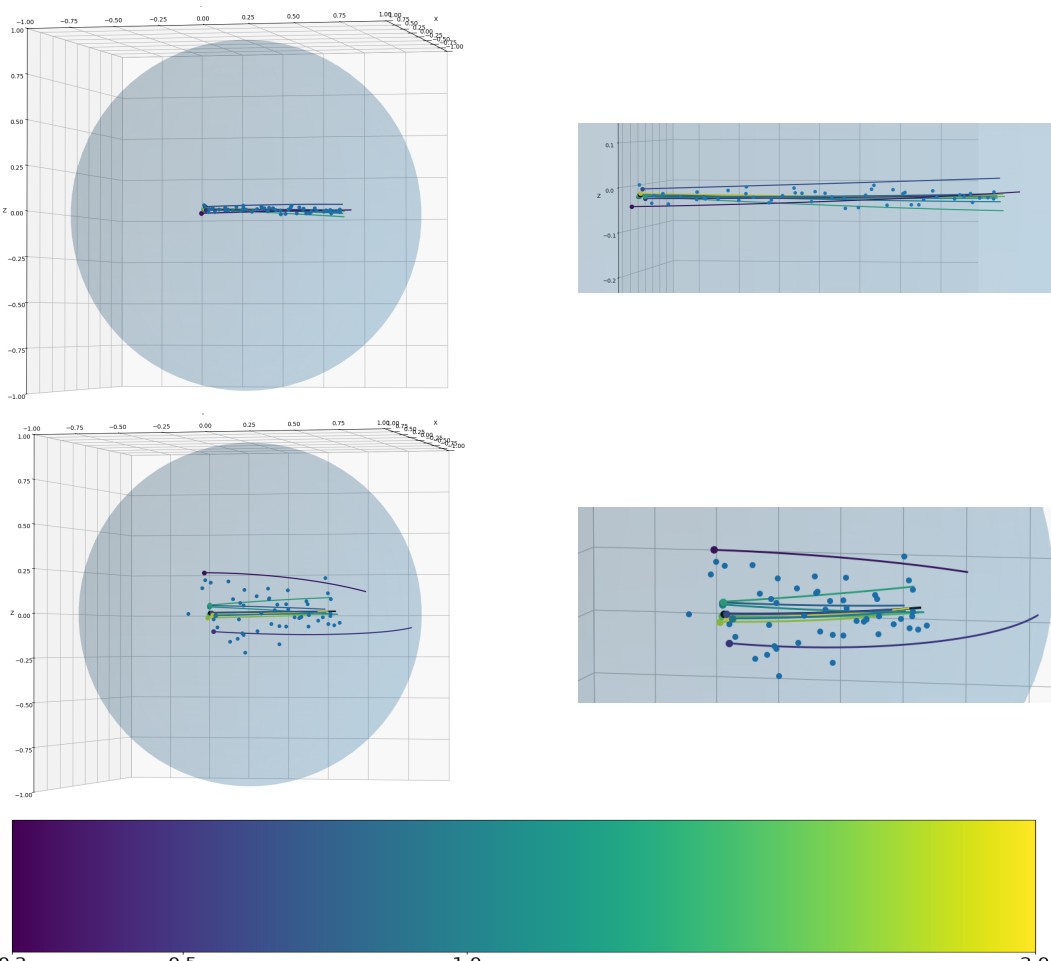

Figure 5: Visual representation of 50 data points (blue dots) along with non-private geodesic and private geodesics for equal allocation of privacy budgets with total budget $\epsilon \in [2.0 - 0.2]$ in ten steps and varying errors $\delta$. Top row: For $\delta = 0.01$, Bottom row: $\delta = 0.1$

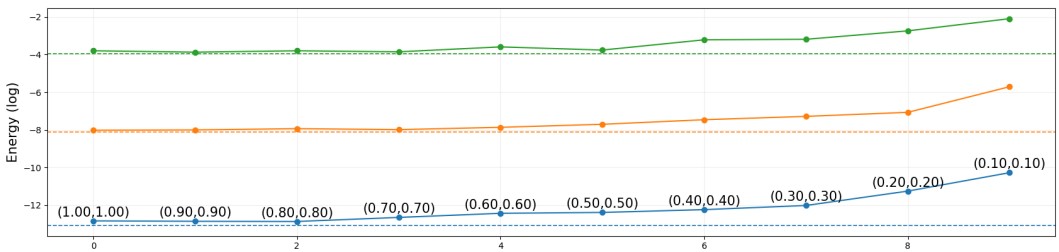

Figure 6: Log-average MSE, $\ln \bar{\mathcal{E}}$, on the unit sphere with 50 data points for $\delta = 0.001$ (blue), $\delta = 0.01$ (orange), and $\delta = 0.1$ (green). Dotted lines indicate the corresponding energies without privatization.

to $\delta = 0.1$, this deviation becomes even more pronounced, with the low–budget geodesics pushed further away from the data.

This behavior is reflected quantitatively in the log-average MSE. Figure 6 shows that, as the noise level increases, the non-private log energy also increases, and for each fixed $\delta$, the log-average MSE exhibits the same trend observed earlier: higher error for smaller total privacy budgets.

# F  SYMMETRIC POSITIVE DEFINITE MATRICES (SPD)

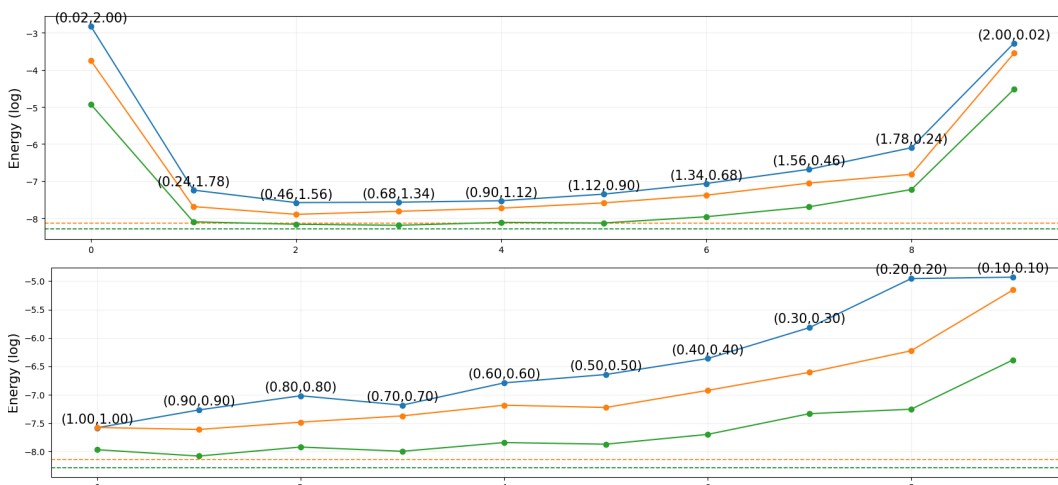

Figure 7: Log average MSE $\ln \bar{\mathcal{E}}$ for 20 (blue), 50(orange), and 100(green) data points on the SPD(2) manifold. Dotted line are the energies without privatisation. Top: unequal budget splits $\epsilon_p \in [0.02, 2.0]$, $\epsilon_v \in [2.0, 0.02]$, total $\epsilon = 2.02$ and Bottom: Equal budget splits with varying total budget $\epsilon \in [0.2, 2.0]$.

We generated synthetic data on the manifold of $2 \times 2$ symmetric positive definite matrices, SPD(2), by sampling points along a geodesic with added noise. Two random matrices $p, q \in \mathrm{SPD}(2)$ were chosen, and the initial tangent vector was set to $v = \mathrm{Log}(p, q)$ using the affine-invariant metric. For covariates $x \in [0, 1]$, clean responses were obtained as $y_{\text{clean}}(x) = \mathrm{Exp}(p, x\, v)$. To model observational variability, each $y_{\text{clean}}(x)$ was perturbed by isotropic Gaussian noise in the tangent space, $\xi \sim \mathcal{N}(0, \sigma^2 I)$, and mapped back as $y(x) = \mathrm{Exp}(y_{\text{clean}}, \xi)$. We used $\sigma = 0.01$, producing small deviations from the underlying geodesic, so that the dataset consists of pairs $\{(x_i, y_i)\}_{i=1}^N$ with $y_i \in \mathrm{SPD}(2)$.

For the SPD(2) manifold, under an affine-invariant metric, the sectional curvature is bounded by $[-\frac{1}{2}, 0]$(Criscitiello and Boumal, 2021) resulting in $\kappa_l = -\frac{1}{2}$. The sensitivities of the footpoint and shooting vector are given by

$$\Delta_p = \frac{2\tau}{n} \cosh\left(2\sqrt{\frac{1}{2}}(\tau_m + \tau)\right).$$

$$\Delta_v = \frac{\tau}{n} \frac{1}{\sqrt{\frac{1}{2}(\tau_m + \tau)}} \sinh\left(2\sqrt{\frac{1}{2}}(\tau_m + \tau)\right).$$

Figure 7 shows the behavior of the average log mean squared error, $\ln \bar{\mathcal{E}}$, for datasets of size 20, 50, and 100, depicted by the blue, orange, and green curves, respectively. The dashed curves in each panel indicate the geodesic log energy corresponding to the non-private regression estimates $(\hat{p}, \hat{v})$.

For each pair of privacy budgets $(\epsilon_p, \epsilon_v)$, we generate 100 private parameter pairs $(\tilde{p}, \tilde{v})$ by sampling 10 candidate footpoints and 10 shooting vectors per footpoint. The top panel reports the case of an unequal budget split, with $\epsilon_p \in [0.02, 2.0]$ and $\epsilon_v \in [2.0, 0.02]$, keeping the total budget fixed at $\epsilon = 2.02$. In this setting, the log error is maximal when either $\epsilon_p$ or $\epsilon_v$ is very small, yielding a parabolic trend reminiscent of what we observed on the sphere and Kendall shape space. The bottom panel presents the equal budget allocation, where the overall privacy budget decreases from 2.0 to 0.2, producing a qualitatively similar pattern.

At smaller budgets, the observed increase in error is expected: tighter privacy constraints induce heavy-tailed sampling distributions for $\tilde{p}$ and $\tilde{v}$, leading to accepted samples that deviate noticeably from the non-private estimates $(\hat{p}, \hat{v})$ and thereby inflate the energy. Although the private estimates yield energies that approach those of the non-private regression, they do not coincide exactly. As the

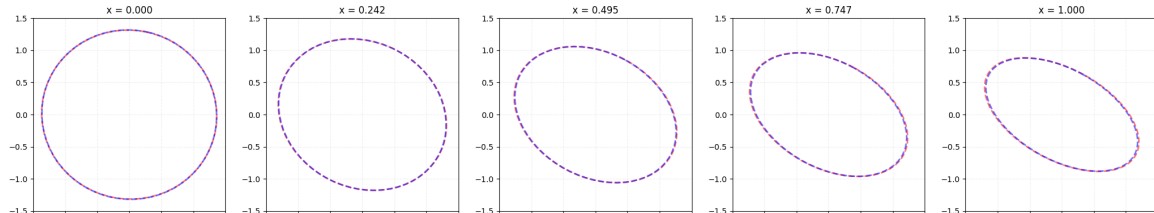

Figure 8: SPD(2) matrices as ellipses for equally spaces covariates. Blue curves denote predictions from the non-private regression parameters, while magenta curves correspond to predictions from the differentially private parameters with $\epsilon_p = \epsilon_v = 0.2$.

sample size grows, the sensitivity reduces, which in turn sharpens the sampling distribution of $(\tilde{p}, \tilde{v})$ around the non-private solution. Consequently, the private energies converge more closely to the geodesic energies, as seen in Figure 7. Across all budget pairs $(\epsilon_p, \epsilon_v)$, the error for 20 data points (blue) exceeds that for 50 points (orange), which is in turn higher than that for 100 points (green).

We know that any element of SPD(2) can be expressed as a real symmetric $2 \times 2$ matrix $y = \begin{pmatrix} a & b \\ b & c \end{pmatrix}$ with $a, c > 0$ and $ac - b^2 > 0$ to ensure positive definiteness. Such a matrix admits an eigendecomposition $y = U\Lambda U^\top$, where $U \in SO(2)$ contains the orthonormal eigenvectors and $\Lambda = \mathrm{diag}(\lambda_1, \lambda_2)$ with $\lambda_1, \lambda_2 > 0$ are the eigenvalues. Geometrically, this representation allows one to view $y$ as defining an ellipse with axes $\sqrt{\lambda_1}$ and $\sqrt{\lambda_2}$ oriented according to $U$.

Figure 8 compares predictions from the non-private estimates (blue) with those from the private estimates using $(\epsilon_p = \epsilon_v = 0.2)$ (magenta) with the ellipse representation. Although representing SPD(2) matrices as ellipses offers an intuitive view of their eigenvalues and orientations, this visualization can be misleading: ellipses that appear similar may be far apart in the Riemannian metric, while visually distinct ones may in fact be close.

## G  VALIDITY OF SENSITIVITY BOUNDS

In this section, we present evidence of the validity of the sensitivity bounds for the unit sphere and SPD(2) found in Section 3 and D. We create artificial adjacent datasets $D, D'$, each of $n$ data points on the given manifold. This is done by generating $n + 1$ data points on the sphere by the same mechanism mentioned at the start of Section 4.1 and F, then removing the first data point for the dataset $D$ and the last data point for the dataset $D'$.

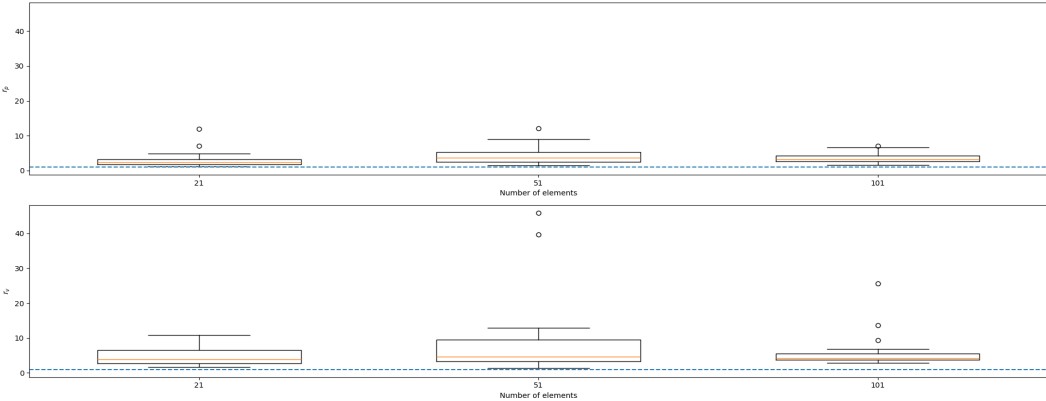

Figure 9: Box plots for ratios $r_p$ and $r_v$ for 20 pairs of adjacent datasets on a unit sphere with $\delta = 0.001$. Left: Box plot of the ratio $r_p = \Delta_p^{thy}/\Delta_p^{exp}$ for $20, 50, 100$ data points on a unit sphere. Right: Box plot of the ratio $r_v = \Delta_v^{thy}/\Delta_v^{exp}$ for $20, 50, 100$ data points on a unit sphere.

Let us define an experimental sensitivity for the footpoint $p$ as $\Delta_p^{exp} = \|\nabla_p E(p; D) - \nabla_p E(p; D')\|$ and for the shooting vector $v$ as $\Delta_v^{exp} = \|\nabla_v E(p; D) - \nabla_v E(p; D')\|$. We call these experimental bounds as the gradients depend on the generated datasets. Next, we define theoretical sensitivity bounds as, $\Delta_p^{thy} = \sup_{D \sim D'} \|\nabla_p E(p; D) - \nabla_p E(p; D')\|$ and $\Delta_v = \sup_{D \sim D'} \|\nabla_v E(v; D) - \nabla_v E(v; D')\|$. From section 4.1 we know that the theoretical bounds for the unit sphere are given by $\Delta_p^{thy} = \Delta_v^{thy} = \frac{2\tau}{n}$. Here we take $\tau = \max_{D,D'} \{\epsilon_j\}$, where $\{\epsilon_j\}$ is the combined set of errors in datasets $D, D'$. To check the validity of our theoretical sensitivity bounds we calculate the ratios $r_p = \frac{\Delta_p^{thy}}{\Delta_p^{exp}}, r_v = \frac{\Delta_v^{thy}}{\Delta_v^{exp}}$. We expect the theoretical bounds to be always greater than the experimental ones as they are defined to be the supremum over all possible adjacent datasets. Thus the ratios $r_p$ and $r_v$ are expected to be greater than one.

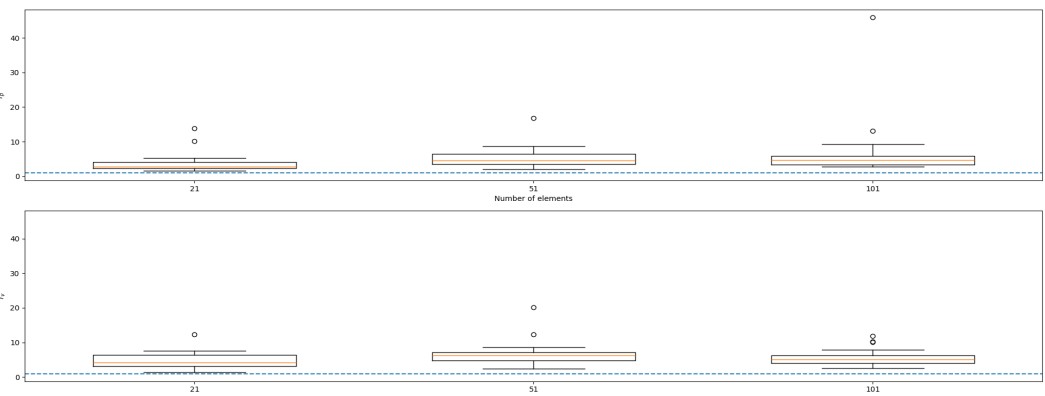

Figure 10: Box plots for ratios $r_p$ and $r_v$ for 20 pairs of adjacent datasets on SPD(2) with $\sigma = 0.01$. Left: Box plot of the ratio $r_p = \Delta_p^{thy}/\Delta_p^{exp}$ for $20, 50, 100$ data points on SPD(2). Right: Box plot of the ratio $r_v = \Delta_v^{thy}/\Delta_v^{exp}$ for $20, 50, 100$ data points on SPD(2).

Figure 9 displays the box plots for the ratios $r_p = \Delta_p^{thy}/\Delta_p^{exp}$ and $r_v = \Delta_v^{thy}/\Delta_v^{exp}$ computed for datasets of size 20, 50, and 100 on the unit sphere. Each box plot is obtained from 20 pairs of adjacent datasets. The ratios are consistently greater than one and concentrated near one, with only a small number of outliers, indicating that the theoretical sensitivity bounds are both valid and tight. Figure 10 presents the corresponding ratios for datasets on the SPD(2) manifold, again using 20 adjacent pairs for each dataset size. The results exhibit the same qualitative behavior, with all ratios exceeding one and remaining close to unity, further confirming the validity and tightness of the bounds in this setting.

