# OpenReview forum: "Differentially Private Geodesic Regression"
_ICLR.cc/2026/Conference — Submitted to ICLR 2026_

### Official Review · Reviewer_Eaui · 2025-10-25

**Soundness:** 2
**Presentation:** 3
**Contribution:** 3
**Rating:** 4
**Confidence:** 3

**Summary:**

The paper studies differentially private (DP) release of geodesic‐regression parameters on Riemannian manifolds. Geodesic regression is parameterized by a footpoint $p\in\mathcal M$ and shooting vector $v\in T_p\mathcal M$; the loss is the mean squared geodesic distance between observations ($y_i\in\mathcal M$) and predictions $\mathrm{Exp}(p,x_i v)$. The authors instantiate the K-Norm Gradient (KNG) exponential mechanism on manifolds to privately sample $\tilde p,\tilde v$, and derive global sensitivity bounds for the gradients of the least-squares energy with respect to $p$ and $v$. Central to the analysis is that sensitivities are controlled by Jacobi fields along the prediction geodesic, hence by the sectional curvature bounds of $\mathcal M$. The main bound (Theorem 3.3) yields $\Delta_p\le \tfrac{2\tau}{n}$ for nonnegative curvature and $\Delta_p\le \tfrac{2\tau}{n}\cosh(2\sqrt{-\kappa_\ell}(\tau_m+\tau))$ for $\kappa_\ell<0$; an analogous bound holds for $\Delta_v$. Experiments on the sphere $S^2$, SPD(2), and Kendall’s planar shape space illustrate error–privacy trade-offs and effects of budget splits $(\varepsilon_p,\varepsilon_v)$.

**Strengths:**

* Clean synthesis of KNG with geodesic regression, yielding curvature-explicit sensitivity bounds; the link through Jacobi fields is elegant and broadly useful.
* Derivations for $\nabla_p E,\nabla_v E$ via adjoints and Jacobi fields, and bounds using Rauch and geodesic-length control are careful and instructive.
* Preliminaries and appendices (sampling on $S^2$; proofs; geometry background) make the paper approachable to the manifold-ML audience.
* A re-usable blueprint for DP mechanisms in manifold regression (beyond geodesics), especially where curvature varies across spaces (sphere, SPD, Kendall).

**Weaknesses:**

1. Using $\tau=\max_i|\varepsilon_i|$ from data to set the mechanism scale breaks DP; results are therefore not privacy-valid as presented. Recommend public/DP bounds (e.g., clipping residuals at a public radius; or releasing a DP (\tilde\tau) with its own budget) and re-running experiments.
2. Missing comparisons to manifold Laplace/Gaussian mechanisms, or to output perturbation of $(\hat p,\hat v)$ under $\mu$-GDP/RDP accounting; also no advanced composition or tighter accounting beyond $\varepsilon_p+\varepsilon_v$.
3. The Riemannian M-H sampler lacks guarantees that approximate sampling error preserves $\varepsilon$-DP; proposal/acceptance choices could leak beyond intended $\varepsilon$. Consider exact samplers where possible, proximal/noisy gradients with DP optimization on manifolds, or rigorous approximation error -> privacy bounds.
4. Sphere experiments are synthetic; SPD(2) appears only in appendix; shape-space study is modest. Include higher-dimensional SPD tasks (e.g., covariance descriptors), larger $n$, and report runtime/acceptance rates vs. budgets.
5. Assumption practicality. Guidance to verify or enforce curvature/radius assumptions in practice is limited; discuss public preprocessing to ensure bounded domains (e.g., projection/clipping under a known atlas).

**Questions:**

1.  Can you replace the data-dependent $\tau=\max_i|\varepsilon_i|$ by either (i) a public bound (domain restriction + clipping) or (ii) a DP estimate $\tilde\tau$ (Laplace/Gaussian on a 1-sensitive statistic) and re-report results? How sensitive are utilities to over-clipping?
2. Why not use $\mu$-GDP/RDP accounting (or advanced composition) to budget $\varepsilon_p,\varepsilon_v$? Could tighter accounting materially change utility?
3. Can you provide bounds on mixing / total variation error and show that sampling error does not compromise privacy? Any chance to adopt DP Riemannian optimization (perturbed gradients on $\mathcal M)$ to avoid MCMC?
4. Please compare against (a) manifold Laplace mechanism applied to $(p,v)$, (b) output perturbation of non-private $(\hat p,\hat v)$, and (c) DP Frechet regression/metric-space regression where applicable, under the same $\varepsilon$.
5. Could you empirically validate the curvature dependence predicted by the bounds (e.g., compare equal-radius synthetic tasks on $S^2$ vs. negatively curved models)?
6. Public feasibility checks. How would practitioners choose public $r,\tau_m$ and ensure $\tau$-closeness without peeking at private data? Any model-selection strategy that is itself DP?

---

> ### Author Response · Authors · 2025-11-19
>
> We thank the reviewer for the very thorough and elaborate review.
>
> W1,Q1,Q6: Thank you for pointing out the weakness in terms of our choice of $\tau$ and $\tau_m$. We address this as a general comment that is available to all.
>
> W2,Q4: The lack of comparisons is largely attributed to the lack of options. The manifold Laplace is indeed a valid contender, but (a) it would require entirely different sensitivity calculations such as $d(p_D,p_{D'})$, which, as we see from our paper, are not clear, and (b) as shown in [1], manifold KNG tends to perform better than Laplace. We will explain our choice of KNG over other mechanisms in the final submission with the reference.
>
> W3: In terms of MCMC, we mainly agree with the reviewer, but we add a note here. On top of the difficulty of finding exact samplers outside of some common distributions, sampling from densities on manifolds is much less explored than its Euclidean counterparts. Further, samplers for densities need to be tailored to specific manifolds and specific metrics; e.g., the Laplace for the SPD matrices with the affine metric is explored in [2], but the sampling scheme for the Laplace would need to be reworked if one were to use SPD matrices with the log-Euclidean metric or Bures-Wasserstein metric.
>
> It is true that MCMC can weaken the guarantees ([1] and [3]). We will be clearer with this and add the cited works, among other related sources. An interesting and important conclusion of the DP-MCMC research is that the privacy guarantees are linked to the run of the algorithm.  As [3] states, "Metropolis-Hastings ... and common variants ... are geometrically ergodic under mild regularity conditions," which thus means a pure DP mechanism turns into a $\epsilon,\delta$ mechanism with $\delta=O(\frac{1}{\text{MCMC chain length}})$. In a real-world scenario, we would release 1 $(\tilde{p},\tilde{v})$ pair; in such a scenario, it would be much simpler to control the privacy weakening. One can have chain length $\rightarrow\infty$ resulting in $\delta\rightarrow 0$.
>
> We note, however, that the intended contribution of this paper is to extend geodesic regression into the DP landscape. We thus agree that our experiments have privacy weaknesses but are robust in
> terms of proof of concept.
>
> w4: In terms of SPD matrices appearing in the appendix, this is largely an issue with space limitations. From our perspective, we did quite a variety of experiments, i.e., 100 pairs of $(\tilde{p},\tilde{v})$ under even and uneven privacy budgets, each at 10 different levels and three different sample sizes, leading to $100\times(2\times10)\times 3=6000$ simulations over 60 settings for each manifold. We agree that experiments over higher-dimensional SPDM would be interesting, but we chose $2\times 2$ as those are the only ones that can be visualized via their eigenvalues.
>
> W5: The assumption has many practical implications. The assumptions are necessary for uniqueness and existence, but also avoids unrealistic scenarios, e.g., imagine fitting a regression on uniform points on the sphere where there is no such meaningful line. We will include guidance on how to verify that the assumptions are satisfied. For instance, one can check the $\max_{i,j} d(y_i,y_j)$ to see if the data is within a sufficiently small ball. The curvature of manifolds is, luckily, well studied and can be computed without knowledge of the data other than its dimension. We will add a note that these quantities are well known and point to references.
>
> Q2: While it is true that there are more advanced ways to account for the total budget, this would not change the utility but rather just allow us to use less budget. This is an interesting point, and admit we are not aware if there exists proofs that show the exponential mechanism satisfies $\mu-$GDP. Converting pure DP to R\'enyi DP is definitely a possibility, but again, this would only change our "total budget" but not the utility. Further, for R\'enyi DP, we'd have to select an $\alpha$ which can lead to questions of its selection.
>
> Q3: This is a very interesting direction which we are currently separately pursuing. Yes, DP Riemannian optimization can likely avoid MCMC sampling, but this leads to other compromises, such as requiring us to work in the $(\epsilon,\delta)$ or $\mu-$GDP framework. Since $(\epsilon,\delta)$ is a "weaker" sense of privacy, this would be the trade-off we'd have to pay.
>
> Q5:
> We did empirically validate our bounds, and we will include some of this validation upon resubmission. We checked the bound for the sensitivity by comparing the theoretical value to empirical value over many simulated adjacent datasets. As the sensitivity captures worst-case scenarios, our investigation showed relatively tightness. It would not be feasible, though, to compare spheres to negatively curved manifolds, as the sphere cannot be represented in one chart, hence now allowing us to embed it into such a space.

---

> > ### Author Response · Authors · 2025-11-19
> >
> > Q6: In addition to what we stated in the general comment, there are many real-world examples where there is available guidance. For instance, if one were to use geodesic regression on the Earth (sphere) to capture migration patterns of birds, the relative variance is known to experts in the field. In a more practical example, the immigration of refugees (again observed on the Earth/sphere), for example, is largely studied and has huge privacy problems.
> >
> >
> > 1. Carlos Soto, Karthik Bharath, Matthew Reimherr, and Aleksandra Slavkovi´c. Shape and structure preserving differential privacy. Advances in Neural Information Processing Systems, 35:24693–24705, 2022
> > 2. Hatem Hajri, Ioana Ilea, Salem Said, Lionel Bombrun, and Yannick Berthoumieu. Riemannian laplace distribution on the space of symmetric positive definite matrices. Entropy, 18(3):98, 2016
> > 3. Jeremy Seeman, Matthew Reimherr, and Aleksandra Slavkovi´c. Exact privacy guarantees for markov chain implementations of the exponential mechanism with artificial atoms. Advances in Neural Information Processing Systems, 34:13125–13136, 2021

---

### Official Review · Reviewer_HY9b · 2025-10-30

**Soundness:** 3
**Presentation:** 3
**Contribution:** 3
**Rating:** 6
**Confidence:** 3

**Summary:**

This paper extends differential privacy techniques to geodesic regression on Riemannian manifolds by adapting the K-Norm Gradient (KNG) mechanism. The authors derive theoretical sensitivity bounds for the regression parameters (footpoint and shooting vector) and demonstrate that these sensitivities depend on Jacobi fields and manifold curvature. The methodology is validated through experiments on three manifolds: the 2-sphere, the space of symmetric positive definite matrices, and Kendall's planar shape space. Results show that privacy-utility tradeoffs behave predictably, with sensitivity decreasing as sample size increases and balanced budget allocation generally outperforming extreme splits.

**Strengths:**

* The paper tackles differential privacy for geodesic regression, a fundamental statistical method for manifold-valued data that appears in sensitive domains like medical imaging (brain tensors, anatomical shapes) and spatial statistics. Given the increasing prevalence of non-Euclidean data in privacy-sensitive applications, providing formal privacy guarantees for this regression framework fills a significant gap in the privacy literature.
* The paper rigorously derives sensitivity bounds (Theorem 3.3) for geodesic regression parameters by connecting them to Jacobi field equations and curvature bounds. This extends previous work on the Fréchet mean and provides a principled framework that generalizes to arbitrary Riemannian manifolds with bounded curvature, making it broadly applicable.

**Weaknesses:**

* The authors acknowledge setting τ = max_i ||ε_i|| which requires examining the data and "indeed violates privacy." This is a fundamental flaw that undermines the practical applicability of the method. While they claim "the concept of our methodology still holds," differential privacy is compromised if data-dependent quantities are used to calibrate noise, and no alternative solution is proposed beyond experimenting with inflated values.

* While the paper examines different budget allocations and sample sizes, it lacks important ablation studies. There is no analysis of MCMC mixing quality, no comparison with baseline methods (e.g., adding noise in ambient space). The number of MCMC samples (m=10 footpoints, 10 vectors each) seems small for reliable posterior characterization.

* The literature review misses several relevant recent contributions that would benefit readers and position the paper more suitably (See [1, 2, 3])


Refs


1) Shape and structure preserving differential privacy.

2) Differentially Private Fréchet Mean on the Manifold of Symmetric Positive Definite (SPD) Matrices with log-Euclidean Metric.

3) Improved Differentially Private Riemannian Optimization: Fast Sampling and Variance Reduction.

**Questions:**

NA

---

> ### Author Response · Authors · 2025-11-19
>
> We thank the reviewer for their diligent work on this review. We address the concerns below.
> \textbf{
> While the paper examines different budget allocations and sample sizes, it lacks important ablation studies. There is no analysis of MCMC mixing quality, no comparison with baseline methods (e.g., adding noise in ambient space). The number of MCMC samples (m=10 footpoints, 10 vectors each) seems small for reliable posterior characterization.}
>
>
>
> \textit{Answer:} In terms of MCMC, we mainly agree with the reviewer, but we add a note here. On top of the difficulty of finding exact samplers, sampling from densities on manifolds is much less explored than its Euclidean counterparts. Further, samplers for densities need to be tailored to specific manifolds and specific metrics; e.g., the Laplace for the SPD matrices with the affine metric is explored in [2], but the sampling scheme for the Laplace would need to be reworked if one were to use SPD matrices with the log-Euclidean metric.
>
> It is indeed true that MCMC can weaken the guarantees ([1] and [3]). We will be clearer with this and add the cited works, among other related sources. An interesting and important conclusion of the DP-MCMC research is that the privacy guarantees are linked to the run of the algorithm.  As [3] states, "Metropolis-Hastings ... and common variants ... are geometrically ergodic under mild regularity conditions," which thus means a pure DP mechanism turns into a $\epsilon,\delta$ mechanism with $\delta=O(\frac{1}{\text{MCMC chain length}})$. In a real-world scenario, we would release 1 $(\tilde{p},\tilde{v})$ pair; in such a scenario, it would be much simpler to control the privacy weakening. One can have chain length $\rightarrow\infty$ resulting in $\delta\rightarrow 0$.
>
> We note, however, that the intended contribution of this paper is to extend geodesic regression into the DP landscape. We thus agree our experiments have privacy weaknesses but are robust in terms of proof of concept.
>
> We would like to emphasize that, in our experiments, we generate $10\times 10=100$ private pairs $(\tilde{p}, \tilde{v})$ for each choice of privacy budget ($\epsilon=(0.2,0.4,\dots,1.8,2.0)$ for even split and $(\epsilon_p,\epsilon_v)={(2.0,0.2),(1.8,0.4),\dots,(0.4,1.8),(0.2,2.0)}$ for uneven split) for each dataset size ($n=20,50,100$) on each manifold.
> This results in a total of $6000$ simulations for each manifold (hence a total of $12000$ not including the shape space example).
> We believe this provides a sufficiently large and diverse set of instances to validate and empirically test our sensitivity bounds. Ideally, more experiments are always more favorable, but as the experiments match the theory, adding more experiments here would largely smoothen the plots rather than add substantive material.
>
>
>
>
> \textbf{The literature review misses several relevant recent contributions that would benefit readers and position the paper more suitably (See [1, 2, 3])}
>
>
>
> \textit{Answer:} We thank the reviewer for the references and will make sure to include them. We note that the first reference is included in the current submission, but did indeed miss the second and third.
>
>
> 1. Andrea Bertazzi, Tim Johnston, Gareth O Roberts, and Alain Durmus. Differential privacy
> guarantees of markov chain monte carlo algorithms. arXiv preprint arXiv:2502.17150, 2025
> 2. Hatem Hajri, Ioana Ilea, Salem Said, Lionel Bombrun, and Yannick Berthoumieu. Rieman-
> nian laplace distribution on the space of symmetric positive definite matrices. Entropy,
> 18(3):98, 2016
> 3. Jeremy Seeman, Matthew Reimherr, and Aleksandra Slavkovi´c. Exact privacy guarantees for
> markov chain implementations of the exponential mechanism with artificial atoms. Advances
> in Neural Information Processing Systems, 34:13125–13136, 2021

---

### Official Review · Reviewer_xHep · 2025-10-31

**Soundness:** 2
**Presentation:** 2
**Contribution:** 2
**Rating:** 2
**Confidence:** 3

**Summary:**

The paper proposes differentially private geodesic regression on Riemannian manifolds via the K-Norm Gradient mechanism, deriving sensitivity bounds that depend on curvature through Jacobi fields. The ICLR version broadens experiments (sphere, SPD, Kendall shape space) and clarifies notation, in comparisons to the previous NeurIPS submission.

**Strengths:**

This paper discusses privacy-preserving learning on manifolds. It is the first work to formulate and analyze geodesic regression under differential privacy, bridging a gap between classical Euclidean differential privacy mechanisms and the geometry of Riemannian manifolds. In our earlier NeurIPS review we identified three main issues: (i) an ill-posed difference in Lemma 3.3 involving adjoints and elements in different tangent spaces; (ii) an incorrect application/direction of the Rauch comparison; and (iii) the privacy implications of the $\tau-$ closeness assumption. While some has improved:
1. **Rauch comparison direction.** The sensitivity analysis now uses a lower curvature bound and obtains cos/cosh-style bounds, which fixes the directionality error we flagged.
2. **Notation and preliminaries.** The manuscript defines the adjoint and states Jacobi initial conditions explicitly, reducing ambiguity.
3. **Sampling/composition.** The sequential treatment of $(p,v)$ via the fiber-bundle view of $TM$ is articulated, clarifying how privacy composition is handled without assuming a global product structure.

However, I still have other concerns as mentioned below.

**Weaknesses:**

## Concerns that remain:
1. The revised proof avoids directly subtracting vectors in different tangent spaces by appealing to operator norms and Jacobi comparisons, but the bound still implicitly depends on choices of parallel transport inside the supremum over adjacent datasets. The paper should add a formal lemma showing transport invariance (or independence from path choices), or otherwise bound uniformly over admissible transports.

2. The assumption that the least-squares geodesic is $\tau$-close (and related $\tau_m$) appears to be determined relative to the confidential data. As written, this is not obviously compatible with DP unless $\tau$ is fixed publicly in advance with projection/truncation, or selected via a DP pre-screening step. This core privacy concern remains unresolved.

3. The NeurIPS version highlighted the Euclidean reduction and compared against standard DP libraries. If the reduction is claimed as a benefit, those Euclidean baselines (or equivalent) should be retained or replicated for completeness.

**Questions:**

see above

---

> ### Author Response · Authors · 2025-11-19
>
> We thank the reviewer for their detailed review and concerns. We address said concerns below.
>
> \textbf{The revised proof avoids directly subtracting vectors in different tangent spaces by appealing to operator norms and Jacobi comparisons, but the bound still implicitly depends on choices of parallel transport inside the supremum over adjacent datasets. The paper should add a formal lemma showing transport invariance (or independence from path choices), or otherwise bound uniformly over admissible transports.}
>
>
> \textit{Answer:} All parallel transport operations in this work are implicitly taken with respect to the Levi–Civita connection of the Riemannian manifold. We will add a statement to this effect. Parallel transport of tangent vectors along a fixed smooth curve is uniquely defined by the Levi--Civita connection of $g$. In our submission, our assumptions state that the data lies in a geodesically convex ball (we will make this more clear) and thus there exists a unique geodesic connecting any two points inside that ball. Therefore, transporting vectors from $T_{p'}M$ to $T_pM$ along the unique minimizing geodesic $\gamma$ yields a uniquely determined linear map
> $$\Gamma_{p' \to p} : T_{p'}M \to T_pM.$$
> Since the Levi--Civita connection is metric compatible and torsion free, parallel transport preserves the Riemannian inner product and hence the induced norm, so $\Gamma_{p' \to p}$ is an isometry. It follows that for any $v \in T_pM$ and $v' \in T_{p'}M$, the difference
> \(
> v - \Gamma_{p' \to p}(v')
> \)
> is unambiguously defined in $T_pM$, and its norm $\| v - \Gamma_{p' \to p}(v') \|_{g(p)}$ does not depend on any choice of path other than the minimizing geodesic guaranteed by the assumption.
>
>
> Lemma: Uniqueness of parallel transport in the data region.
>
>
> Suppose $p,p' \in B_r(m_0)$. Then there exists a unique minimizing geodesic $\gamma:[0,1]\to M$ joining $p$ and $p'$, and parallel transport along $\gamma$ with respect to the Levi--Civita connection defines a unique linear isometry
> \[
> \Gamma_{p' \to p} \colon T_{p'}M \longrightarrow T_pM.
> \]
> In particular, for any $v \in T_pM$ and $v' \in T_{p'}M$, the quantity
> \[
> \bigl\| v - \Gamma_{p' \to p}(v') \bigr\|_{g(p)}
> \]
> is well-defined, independent of any path choices, and depends only on the pair $(p,p')$.
>
>
> \begin{proof}
> By the Hopf--Rinow theorem, completeness of $(M,g)$ implies that any two points in $M$ can be joined by at least one minimizing geodesic. Under assumption that data is inside a strongly geodesically convex (as we avoid cut locus and $r<inj(m_0)$) ball $B_r(m_0)$: for any $p,p'\in B_r(m_0)$ there exists a \emph{unique} minimizing geodesic $\gamma:[0,1]\to M$ with $\gamma(0)=p$ and $\gamma(1)=p'$, and this geodesic remains inside $B_r(m_0)$.
>
> \end{proof}
>
> \textbf{The NeurIPS version highlighted the Euclidean reduction and compared against standard DP libraries. If the reduction is claimed as a benefit, those Euclidean baselines (or equivalent) should be retained or replicated for completeness.}
>
>
> \textit{Answer:} After careful examination, we realized that we had inadvertently used incorrect DP energy values from their models. As a result, comparing our DP energies to theirs would not be meaningful or fair. While linear regression is an immediate special case of our framework, we chose to omit it for now because, without a reliable and correct comparative study, including only the linear regression case does not substantively strengthen the paper.

---

> ### Comment · Reviewer_xHep · 2025-11-28
>
> Thank you for your response. Your clarifications have addressed my main concerns. I will raise the score accordingly.

---

### Official Review · Reviewer_KD4m · 2025-10-31

**Soundness:** 3
**Presentation:** 3
**Contribution:** 2
**Rating:** 6
**Confidence:** 2

**Summary:**

The paper studies differentially private geodesic regression, which can be viewed as linear regression generalized to cases where the output lives on a curved space rather than in ordinary Euclidean space. It proposes releasing the fitted p and v with the K-Norm Gradient mechanism, adapted to work directly on a Riemannian manifold. The key technical step is to bound how much the loss gradient can change when one data point changes, using Jacobi fields so that the required noise depends on curvature and on a radius bound for the data. The method samples from the resulting target distribution with a Riemannian MCMC routine and reports results on the sphere and on Euclidean space.

**Strengths:**

Overall, this paper is technically sound. I feel like the major contribution is to apply KNG with tight analysis on the geometry and sensitivity.

**Weaknesses:**

The authors implement a Riemannian random-walk MH sampler to draw from the KNG density. Because DP is guaranteed for the target distribution, numerical sampling error can in principle weaken guarantees.

**Questions:**

The experiments empirically choose 10 different epsilon_p and epsilon_v pairs. Is there any analytical guidance or asymptotic criterion suggesting an optimal split under curvature? My concern is that parameter tunning would also cost privacy budget in practice.

---

> ### Author Response · Authors · 2025-11-19
>
> We thank the reviewer for the attention to detail and their review. We address the concerns below.
>
> \textbf{The authors implement a Riemannian random-walk MH sampler to draw from the KNG density. Because DP is guaranteed for the target distribution, numerical sampling error can in principle weaken guarantees:}
>
>
> \textit{Answer:} In terms of MCMC, we mainly agree with the reviewer, but we add a note here. On top of the difficulty of finding exact samplers, sampling from densities on manifolds is much less explored than its Euclidean counterparts. Further, samplers for densities need to be tailored to specific manifolds and specific metrics; e.g., the Laplace for the SPD matrices with the affine metric is explored in [2], but the sampling scheme for the Laplace would need to be reworked if one were to use SPD matrices with the log-Euclidean metric.
>
> It is indeed true that MCMC can weaken the guarantees ([1] and [3]). We will be clearer with this and add the cited works, among other related sources. An interesting and important conclusion of the DP-MCMC research is that the privacy guarantees are linked to the run of the algorithm.  As [3] states, "Metropolis-Hastings ... and common variants ... are geometrically ergodic under mild regularity conditions," which thus means a pure DP mechanism turns into a $\epsilon,\delta$ mechanism with $\delta=O(\frac{1}{\text{MCMC chain length}})$. In a real-world scenario, we would release 1 $(\tilde{p},\tilde{v})$ pair; in such a scenario, it would be much simpler to control the privacy weakening. One can have chain length $\rightarrow\infty$ resulting in $\delta\rightarrow 0$.
>
> We note, however, that the intended contribution of this paper is to extend geodesic regression into the DP landscape. We thus agree that our experiments have privacy weaknesses but are robust in terms of proof of concept.
>
> \textbf{The experiments empirically choose 10 different $\epsilon_p$ and $\epsilon_v$ pairs. Is there any analytical guidance or asymptotic criterion suggesting an optimal split under curvature? My concern is that parameter tunning would also cost privacy budget in practice.}
>
>
> \textit{Answer:} We do not have any theoretical guarantees on the optimal split, unfortunately. However, taking into account that $p$ and $v$ have the same dimension and the empirical results, it seems that an even split always performs well. The concern about tuning the parameters costing budget is valid, but practically, one would only sample from the density once.
>
>
> 1. Andrea Bertazzi, Tim Johnston, Gareth O Roberts, and Alain Durmus. Differential privacy guarantees of markov chain monte carlo algorithms. arXiv preprint arXiv:2502.17150, 2025
> 2. Hatem Hajri, Ioana Ilea, Salem Said, Lionel Bombrun, and Yannick Berthoumieu. Riemannian laplace distribution on the space of symmetric positive definite matrices. Entropy,
> 18(3):98, 2016
> 3. Jeremy Seeman, Matthew Reimherr, and Aleksandra Slavkovi´c. Exact privacy guarantees for markov chain implementations of the exponential mechanism with artificial atoms. Advances in Neural Information Processing Systems, 34:13125–13136, 2021

---

> > ### Comment · Reviewer_KD4m · 2025-11-26
> >
> > Thank you for your response. Your clarifications address my main concerns. I agree that extending DP to geodesic regression is the core contribution and that the experiments provide a robust proof-of-concept.

---

### Author Response · Authors · 2025-11-19

Again, we thank all the reviewers for all their diligent work. We wanted to address the concern with $\tau$ and $\tau_m$ as a whole as all the reviewers shared similar views.

In DP, a priori bounding data is a delicate procedure. For instance, consider a dataset $D$ of ages of humans, it might be reasonable to assume the ages are bounded between 0 to 122. This upper bound, however, can theoretically be exceeded/violated. Similarly, if one had more information, such as "ages of humans with Alzheimer's," one might be inclined to use a lower bound of approximately 65, as this is reasonable. Still, this can (and has) been violated (early onset Alzheimer's can arise in one's 30s). Each of these settings requires some public knowledge. While we can use public information to guide how we bound the data, these bounds are still subject to possible violations.

We agree with the reviewers, our choice of $\tau$ and $\tau_m$ can be improved. As reviewer \textbf{Eaui} rightly pointed out (or rather asked), we can replace our $\tau$ and $\tau_m$ with a sanitised version, say $\tilde{\tau}, \tilde{\tau}_m$. However, we would like to point out that since $\sigma\propto \tau$, replacing $\tau$ with $\tilde{\tau}>\tau$ and $\tau_m$ with $\tilde{\tau}_m>\tau_m$  we'd simply get a slightly larger noise scale. Further, since $\sigma\propto \tau/\epsilon$ "increasing $\tau$" behaves similarly to "decreasing $\epsilon$", which is one reason we included experiments (Fig 1, Fig 2, Fig 5, Fig 6, and Fig 7) where we examined the behavior of our methodology under varying $\epsilon$. Lastly, even in the case where we have a private $\tau$, just as with the earlier Alzheimer's example, it may still be possible that $\tilde{\tau}$ is not large enough. This is a persistent problem in DP.

Seeing as we generally agree with the reviewers, we will add a longer discussion on the choice of $\tau$ and $\tau_m$ and guidance on how it can be achieved in a private manner. As noted earlier, though, our goal of this paper is to extend geodesic regression into the DP landscape. We believe we have achieved this.

---

### Author Response · Authors · 2025-12-02

Area Chairs and Senior chairs,

As papers have been shuffled due to a data leak, we summarize some of the discussion for the new chairs. There are two concerns that a few reviewers had which we summarize first.
* **The use of MCMC:** Three reviewers mentioned our use of MCMC as a concern. In our rebuttals, we appealed to the literature of DP-MCMC to show that while one needs to be cognizant of the potential privacy leakage, our main goal was to extend geodesic regression to the DP landscape. As our methodology is for Riemannian manifolds, sampling methods are rarely available in closed form and even more rarely are they exact samplers. That being said, while the concern is valid, it is mainly a concern for simulations as practically one can make $\delta$ approach 0.
* **The choice of tau:** Three reviewers mentioned our non-private tau as a major concern. As a response, we wrote a general comment. We agree that the tau we use is not private; however, we noted that extending regression was the main goal here. As we mentioned in the overall comment, sanitizing this error bound is a persistent problem in DP. We mentioned we will add guidance on how one would make this tau private upon resubmission.

The reviewers stated that our responses to the above satisfactorily addressed their concerns. We summarize individual reviewer concerns below.
* **Reviewer KD4m:** The major concerns this reviewer had were 1. our use of MCMC for a pure-DP mechanism and 2. guidance on how to split the privacy budget between the two parameters. We mention our resolution for 1. above. In terms of point 2, we referred to our empirical results to show how an even split of the budget always performs well. For both of these concerns, the reviewer was concerned about robustness and we point out that our experiments covered a wide range of scenarios which demonstrated the requested robustness. **Reviewer gave an initial score of 6 and on Nov 26, they mentioned we addressed their main concerns. The score remained the same, but given we addressed the concerns and ICLR locked the score system, we are not certain if there was an intention to increase the score.**
* **Reviewer xHep** The major concerns of this reviewer are on 1. the use of parallel transport in our proof, 2. the lack of a Euclidean counterpart, and 3. the choice of tau.
    In terms of 1, we included a lemma in our comments on the use of our parallel transport, demonstrating the validity of our proof. In terms of 2. this was a reference to a previous NeurIPS submission which we excluded, as a NeurIPS reviewer mentioned that we may not have utilized the blackboxes appropriately and thus a fair comparison was not possible. We thus excluded a Euclidean application; however our framework is applicable to such a setting. In terms of 3 we addressed as above.
    **The reviewer had an initial score of 2 and they mentioned in their comment that we addressed and that they ``will raise the score accordingly." Unfortunately, the reviewer did not have the opportunity to raise the score before the system was locked due to the leak.**
* **Reviewer HY9b** The major concerns of this reviewer are 1. The use of MCMC and 2. the number of simulations. Point 1. we addressed as in above. For point 2, we added some clarification as to the number of experiments we did. As some experiments are in the appendix, due to the limitation of space, they were perhaps overlooked. The reviewer did not have a chance to respond to our rebuttal. **The initial score of this reviewer is 6.**

* **Reviewer Eaui** The review had many valid questions and pointed at some weaknesses. Among these, the major concerns seemed to be 1. lack of comparisons, 2. the use of MCMC, 3. the number of experiments. In our response, we included in-depth responses to their questions and the reviewer mentioned we addressed their concerns.
   **The reviewer had an initial score of 4, which increased to 6 as we had addressed their major concerns during the rebuttal period.**


**Summary:** We had initial scores of 6,6,4,2 which updated to 6,6,6,2. Of the updated scores, the 2 was going to be increased (as we addressed all their concerns), and possibly a 6 was going to be increased, but we are not certain of the latter.  We have submitted a revised version of the paper with the promised changes. We have included appendix sections with the discussions of tau, elaborated on the MCMC, and included notes on parallel transport. Further we have added some plots on the validity of our bounds as proposed by reviewer Eaui. We believe these changes improve the paper as suggested by the reviewers.

---

### Meta-Review · Area_Chair_1LQ4 · 2025-12-22

**Summary:**

This paper proposes a differentially private mechanism for geodesic regression on manifolds via a KNG-style release of the footpoint and shooting vector, with curvature-explicit sensitivity analysis derived using Jacobi fields. Reviewers generally appreciate the contributions as novel and technically motivated. However, the decision-critical concerns are about: (i) the sampling uses a data-dependent error bound, which the paper acknowledges is not private, undermining the DP claim as executed, and (ii) reliance on approximate MCMC sampling further weakens the privacy story without standard privacy diagnostics. In addition, the paper should include comparisons against other private baselines—both (a) private algorithms designed directly on manifolds (e.g., manifold Laplace/Gaussian mechanisms, manifold private optimization, output perturbation variants) and (b) Euclidean DP methods combined with projection back to the manifold, together with utility–privacy tradeoff curves. In light of the above concerns, I could not recommend acceptance at current stage.

**Reviewer Concerns:**

The concerns that remain outstanding are (1) Impact of MCMC sampling on the privacy (2) Sampling relies on data-dependent error bound (3) comparisons to private baselines. For the first two, the authors acknowledged in the rebuttal and added brief discussions as to how they may be addressed. This is insufficient in my view as they form the core components of the paper. For the last point, the authors did not provide comparisons.

**Reviewer Scores:**

Reviewer KD4m appears not an expert given their low confidence score and remain borderline. Reviewer xHep may increase the score but still remain borderline reject given the insufficient address of his concerns. Reviewer HY9b also remains borderline. Reviewer Eaui is likely to increase the score but still maintain borderline. Overall, the paper is on the borderline.

---

### Decision · Program_Chairs · 2026-01-26

Reject